# Paleolithic occupation of arid Central Asia in the Middle Pleistocene

Emma M. Finestone[1,2]*, Paul S. Breeze[3], Sebastian F. M. Breitenbach[4], Nick Drake[2,3], Laura Bergmann[5], Farhod Maksudov[6], Akmal Muhammadiyev[6], Pete Scott[7], Yanjun Cai[8], Arina M. Khatsenovich[9], Evgeny P. Rybin[9], Gernot Nehrke[10], Nicole Boivin[2,11,12,13], Michael Petraglia[2,11,14,15]

1 Department of Anthropology, The Cleveland Museum of Natural History, Cleveland, OH, United States of America, 2 Department of Archaeology, Max Planck Institute for the Science of Human History, Jena, Germany, 3 Department of Geography, Kings College London, London, United Kingdom, 4 Department of Geography and Environmental Sciences, Northumbria University, Newcastle upon Tyne, United Kingdom, 5 Department of Physical Geography, Catholic University of Eichstätt-Ingolstadt, Eichstätt, Germany, 6 National Center of Archaeology, Uzbekistan Academy of Sciences, Tashkent, Uzbekistan, 7 Centre for Microscopy, Characterisation and Analysis, The University of Western Australia, Perth, Australia, 8 Institute of Global Environmental Change, Xi'an Jiaotong University, Xi'an, China, 9 Institute of Archaeology and Ethnography of the Siberian Branch of the Russian Academy of Sciences, Novosibirsk, Russia, 10 Alfred-Wegener-Institut Helmholtz-Zentrum für Polar- und Meeresforschung, Bremerhaven, Germany, 11 School of Social Science, University of Queensland, Brisbane, Australia, 12 Department of Anthropology and Archaeology, University of Calgary, Calgary, Canada, 13 Department of Anthropology, National Museum of Natural History, Smithsonian Institution, Washington, DC, United States of America, 14 Human Origins Program, National Museum of Natural History, Smithsonian Institution, Washington, DC, United States of America, 15 Australian Research Centre for Human Evolution, Griffith University, Brisbane, Australia

* finestone@shh.mpg.de

**Data Availability Statement:** All relevant data are within the paper and its Supporting Information files.

## Abstract

Central Asia is positioned at a crossroads linking several zones important to hominin dispersal during the Middle Pleistocene. However, the scarcity of stratified and dated archaeological material and paleoclimate records makes it difficult to understand dispersal and occupation dynamics during this time period, especially in arid zones. Here we compile and analyze paleoclimatic and archaeological data from Pleistocene Central Asia, including examination of a new layer-counted speleothem-based multiproxy record of hydrological changes in southern Uzbekistan at the end of MIS 11. Our findings indicate that Lower Palaeolithic sites in the steppe, semi-arid, and desert zones of Central Asia may have served as key areas for the dispersal of hominins into Eurasia during the Middle Pleistocene. In agreement with previous studies, we find that bifaces occur across these zones at higher latitudes and in lower altitudes relative to the other Paleolithic assemblages. We argue that arid Central Asia would have been intermittently habitable during the Middle Pleistocene when long warm interglacial phases coincided with periods when the Caspian Sea was experiencing consistently high water levels, resulting in greater moisture availability and more temperate conditions in otherwise arid regions. During periodic intervals in the Middle Pleistocene, the local environment of arid Central Asia was likely a favorable habitat for paleolithic hominins and was frequented by Lower Paleolithic toolmakers producing bifaces.

**Funding:** This work was funded by the Max Planck Society: https://www.shh.mpg.de/en. Research by PSB was funded by the Leverhulme Trust (ECF-2019-538). The funders had no role in study design, data collection and analysis, decision to publish, or preparation of the manuscript.

**Competing interests:** The authors have decades that no competing interests exist.

# Introduction

Central Asia is situated at a crossroad that links east and north Asia with Europe and the Levant. This region is fundamental to questions of early hominin dispersals because of its position at the gateway between key regions where at least two Middle Pleistocene hominin species are known to have interacted [1]. Subsequently our own species also moved through these regions [2, 3], with the routes and timings of its initial dispersals remaining debated. Despite the importance of Central Asia for understanding the spatial and temporal patterning of hominin occupations in Eurasia, however, our knowledge of hominin activity in this vast and diverse landscape is disproportionately limited when compared with other regions on the continent.

Because of the absence of dated and stratified Lower Paleolithic sites in Central Asia, most studies have focused on the region's Middle and Upper Paleolithic occupation. Currently, available evidence indicates that the Pamir, Tian Shan, and Altai mountains served as corridors of occupation and movement for populations of multiple hominin species through the Late Middle and Late Pleistocene, including Denisovans, Neanderthals, and modern humans [4]. Hominins occupied Central Asia consistently through the Late Pleistocene, even throughout periods of climatic downturn during the coldest episodes of the Last Glacial Period [5–8]. However, the initial occupation of Central Asia and the role the low and mid-altitude plains played in dispersal and occupation remain poorly understood and contentious.

The early colonization of Central Asia has been previously reviewed most notably by Ranov and Davis [9], Davis and Ranov [10], Vishnyatsky [11], Derevianko [12], and Glantz [13]. However, understanding the environmental dynamics of these regions is made difficult by the lack of well-dated paleoclimate records and the paucity of stratified lithic assemblages from Pleistocene arid Central Asia. These limitations remain the most important hurdle for the systematic study of the Lower Paleolithic in this region.

## Climate of arid Central Asia

Central Asia encompasses the former Soviet republics of Kazakhstan, Uzbekistan, Turkmenistan, Tajikistan, and Kyrgyzstan (Fig 1). The interior of Central Asia sits at mid- and low-altitudes and is geographically diverse, being characterized by both tectonic activity and continentality. The vast majority of the region consists of a relatively flat desert and semi-desert plain (i.e., the Turan Depression) bounded in the south by the Tien Shan, Pamir, and Hindu Kush mountains, to the southwest by the Kopetdag, Alborz and Zagros mountains, to the northwest by the Ural Mountains, and to the east by the Altai Mountains. Its western border is in part defined by the Caspian Sea and its northern one by the Siberian plains (Fig 1).

The climate of the mountains is largely controlled by the westerlies bringing moisture to this region [17]. However, the large distance from the global ocean and the topography of the surrounding mountains isolate the lowlands, resulting in dry continental climate conditions, which have produced the Karakum and the Kyzylkum Deserts. Consequently, the water supply in the region is largely delivered by winter precipitation (i.e., rain and snow) at high altitudes, and snow/glacier melt during the summer.

Drainage from the mountains into the Aral and Caspian Seas provides the primary water supply throughout the arid Central Asian lowlands. However, the paleoclimatic history of the Aral Sea shows it as a transient water body, only forming a large lake when the Amu Darya River flows into it, as it does today. Yet this river has switched between the Caspian and Aral Seas numerous times during the Pleistocene, causing the lake to fluctuate substantially [18]. The size of the Caspian Sea has also fluctuated greatly during the Pleistocene [19, 20]. This is partly due to it gaining and losing waters from the switching of the Amu Darya River.

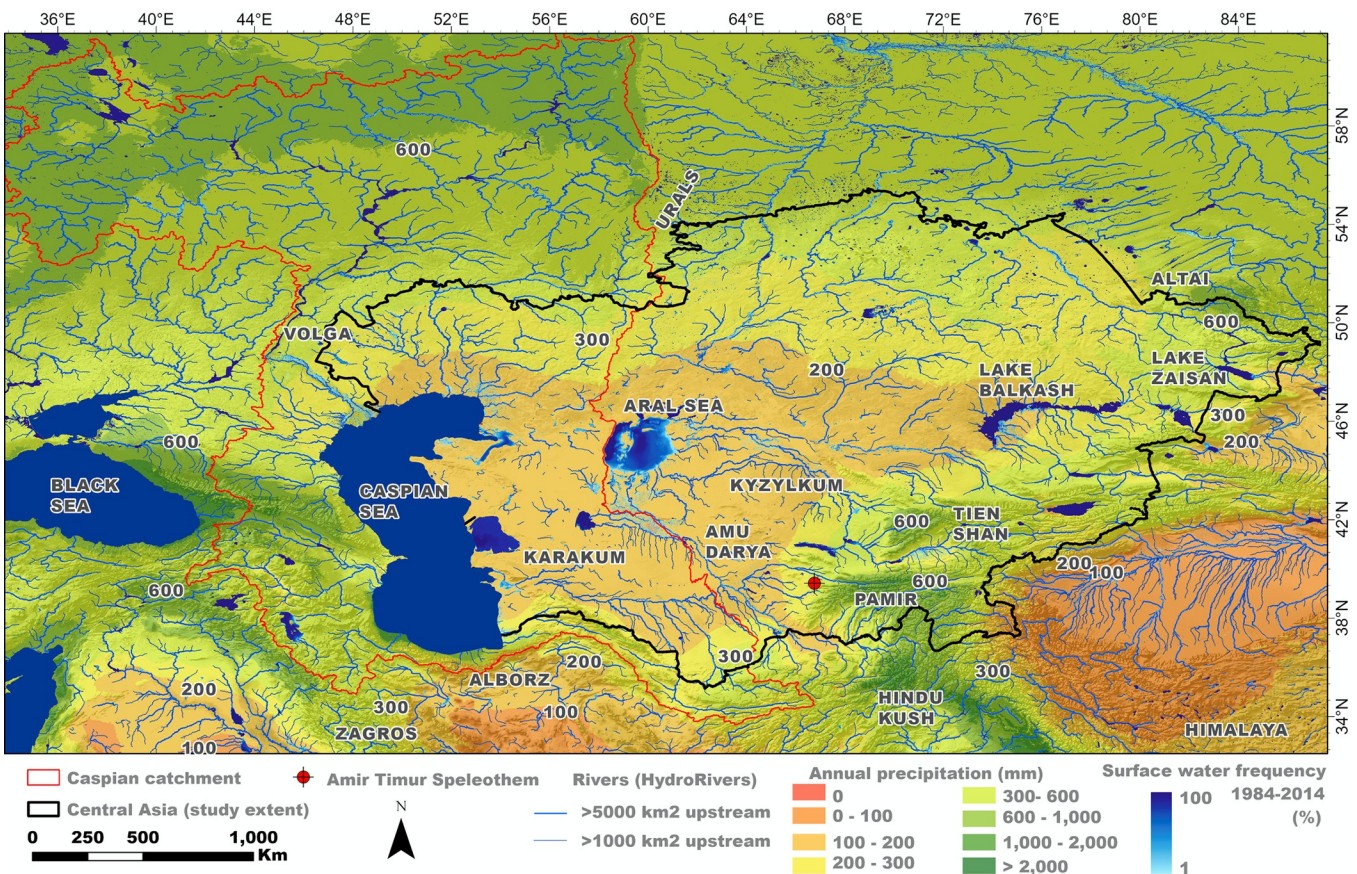

**Fig 1. The study region.** A map of Central Asia derived by the authors using data for annual precipitation (Worldclim [14]), surface water [15], and rivers (Hydrosheds [16]). The study region (encircled with black line) is shown with the region of Caspian catchment (enclosed with red line). Key mountain ranges, deserts, and bodies of water are labeled. The location of Amir Timur Cave that hosted stalagmite S-12-4 analyzed in this study is shown as red circle.

However, given the fact that the rivers feeding the Caspian Sea span so many different climates and altitudinal zones (Fig 1), it is clear that a multitude of factors would have caused changes to the size of the lake during the Pleistocene. The main river feeding the Caspian Sea today is the Volga. It has an extensive catchment feeding in from the north, with its headwaters as far north as Moscow. Thus, changes in moisture availability and ice sheet fluctuations in this region will cause lake level variations over time, with particularly high discharge occurring at the termination of glacial periods when the Scandinavian Ice sheet rapidly melts [19].

Though the mountains of Central Asia provide water to the Caspian and Aral Seas, evaporation from these waterbodies in turn provides an important moisture source for the precipitation in the mountains. For example, isotopic studies of ice cores from the Tien Shan and Altai Mountains suggest that about a third of the Altai precipitation is derived from the Aral and Caspian Seas, whereas 87% of the precipitation in the Tien Shan Mountains is derived from the Aral, Caspian, Mediterranean and Black Seas [21]. Thus, the fluctuations in the size of the Pontocaspian seas will cause fluctuations in the amount of precipitation within Central Asia.

The climatic trends that resulted in the modern arid climate of Central Asia took root in the Early Pleistocene [22, 23]. At the beginning of the Pleistocene, the climate was semi-arid, but warmer and wetter than later periods [22]. Annual rainfall and temperature steadily decreased through the Middle Pleistocene [23]. This trend is tied in part to the Middle Pleistocene Transition (MPT, 1250–700 ka), during which the high amplitude periodic 100-kyr glacial-

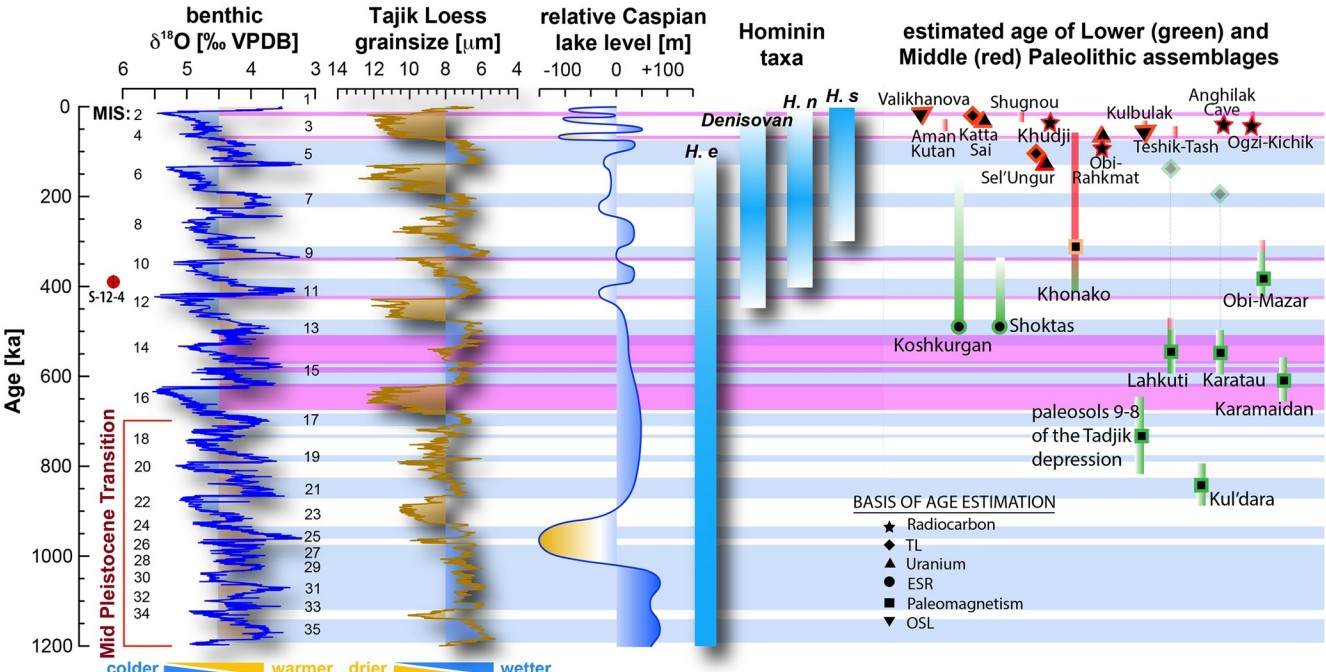

**Fig 2. Climate variables and evidence of hominin occupation in Central Asia through time.** Global oxygen isotope records with numbered MIS interglacials (CENOGRID dataset) [29], the Tajik loess record [26], and relative Caspian Sea level variation [19]. Blue shading indicates wet intervals in the Tajik loess record and pink shading indicates intervals of Caspian outflow into the Black Sea [19]. All known hominin taxa from Central Asia during the Pleistocene (*Homo erectus* [H.e.], Denisovans, *Homo neanderthalensis* [H.n.], *Homo sapiens* [H.s.]) are displayed alongside the suggested chronology for Lower (green) and Middle (red) Paleolithic assemblages. Symbols indicate the basis of age estimations and assemblages without symbol represent dates previously proposed in the literature but are unverified/not tested by dating methods.

interglacial cycles emerged (Fig 2) [24, 25], promoting dryland environments throughout the lowland in the mid-altitude of Central Asia, particularly during glacial periods [25]. Notwithstanding this, the expansion of the Karakum and Kyzylkum deserts likely took place gradually in the Middle Pleistocene, as indicated by a progressive increase in loess deposits [25–28]. However, after the MPT these loess records show significantly increased changes in grain size that are largely coincident with glacial interglacial cycles, indicating that there were significant variations in aridity throughout this period, with interglacials generally being wetter and glacials being drier. Thus overall, the intensification of glacial-interglacial cycles contributed to the aridification of Central Asia through the Middle Pleistocene.

While loess profiles inform on glacial-interglacial environmental changes with (at best) centennial to multi-decadal resolution, speleothem-based proxy records can allow detailed insights into past hydrology and environmental variability even at seasonal scale. Well-dated and highly resolved reconstructions of Pleistocene regional and local environmental changes are extremely scarce in Central Asia, however. Regional reconstructions of Pleistocene climatic conditions rely mostly on loess sequences [26] and only two speleothem records from Tonnel'-naya and Kesang caves [30–32]. This dearth of high-resolution data on past hydrological and thermal conditions severely limits our understanding of human activity in this highly dynamic environment.

## The timing of the earliest occupation of Central Asia

Hominins first dispersed out of Africa and into Asia approximately two million years ago. This is evidenced by the existence of Lower Paleolithic toolkits in China by 2.1 Ma [33] and fossil

remains in Dmanisi, Georgia, dated to around (ca.) 1.8 Ma [34–36]. Hominin remains are present prior to 1 Ma in Georgia (i.e., Dmanisi ca. 1.8 Ma), Turkey (Kocabaş, ca. 1.3–1.1 Ma) [37], China (Gongwangling, ca. 1.63 Ma) [38], and Indonesia (Java, ca. 1.6–1.5 Ma) [39]. The lack of hominin remains or stone tools prior to 1 Ma in Central Asia may suggest that early hominin dispersals into East Asia bypassed Central Asia completely and later entered Central Asia either through an east-to-west route, or in subsequent migration events from Africa north-eastward. However, Early Pleistocene stratified deposits are sparce and largely unexplored in Central Asia. It is possible that hominins initially colonialized and dispersed across Central Asia during Early Pleistocene migrations to northern China, but that evidence is currently lacking because of an absence of systematic investigations of this time interval.

Although the archaeological record is sparse and dating difficult, some evidence suggests that between 1.0–0.8 Ma hominins inhabited Central Asia and produced Lower Paleolithic toolkits. The oldest age for any Paleolithic locality in the region has been proposed for an assemblage of Mode 1 lithics from eastern Kopetdag, in the valley of the River Keshefrud [40], in southwestern Central Asia. While the proposed geological age for this assemblage is between 1 and 0.8 Ma [40], the basis for this date is unknown and unsubstantiated [11]. Another Lower Paleolithic assemblage identified in Yangadzha, Turkmenistan, may also date to the Middle Pleistocene based on correlation with similar technocomplexes in the Caucasus [10]. However, the lack of stratified, datable archaeological or fossil material from Turkmenistan makes it impossible to verify the arrival of hominins in the Kopetdag region in the Early or Middle Pleistocene.

More convincing evidence comes with the oldest known stratified localities in Central Asia, identified in loess deposits in Tajikistan. A reliable date is associated with stone tools from the Kul'dara location in the Tajik Depression [41, 42]. Artifacts from Kul'dara are associated with pedocomplexes (PC) 12 and 11 [41, 42] and were excavated from strata between the Brunhes/Matuyama boundary and the Jaramillo subchron, indicating an age between 900–800 Ka (reviewed in [11]). Kul'dara is the oldest assemblage in this region, although other Middle Pleistocene localities with stratigraphic contexts are known from the Tajik loess deposits [10]. Isolated artifacts recovered between PC 11 and 6 [42], including finds from Karamaidan (PC 6; [43]) are likely older than 600 ka [11]. Larger collections of artifacts resembling similar but later forms of Kul'dara industries were recovered at Karatau (PC 6), Lakhuti (PC 5), Obi-Mazar (PC 4), and Khonako (PC 4; [44–47]). Dating frameworks indicated that Karatau and Lakhuti were formed approximately 500 to 600 ka [23, 48].

Lower Paleolithic toolkits found in Kazakhstan have also been attributed to the Middle Pleistocene. Two excavated localities from the South-Kazakhstan Province in the foothills of the Karatau mountains region (Koshkurgan and Shoktas) have each yielded hundreds of artifacts. The oldest of these deposits were dated by electron spin resonance (ESR) to ca. 500 ka [49]. However, these dates are accepted with caution and it is possible that the Koshkurgan-Shoktas industry could be attributed to the Middle Paleolithic. In eastern Kazakhstan, Lower Paleolithic artifacts are mixed into deflated assemblages along the Zaisan Basin (e.g., Kuruchum) [50, 51]. At several locations, artifacts have been found eroding from what are estimated to be Early/Middle Pleistocene deposits, and the patination on the most archaic of these artifacts indicates warm and relatively humid conditions [52].

Aside from the Tajik loess deposits and the Koshkurgan-Shoktas complex, Lower Paleolithic finds in Central Asia provide very little basis for dating. However, it is worth noting that the handful of Lower Paleolithic sites with established dates fall within the Middle Pleistocene or older, suggesting that the majority of Lower Paleolithic artifacts may have been produced during this time period. In any case, by the Middle Pleistocene, hominins were clearly

inhabiting Central Asia (Fig 2). However, areas of Lower Paleolithic occupation were likely inhabited intermittently and primarily during warm and relatively humid interglacial periods [44, 53].

## Features of the lower paleolithic of Central Asia

The Lower Paleolithic of Central Asia presents a great deal of variability in tool form and assemblage composition. Lithic assemblages are generally attributed to the Lower Paleolithic based on the presence of core-and-flake industries or large Acheulean bifaces. Glantz [13] divides these diverse assemblages into four industrial complexes: micro artifacts, core-and-flake industries, pebble culture, and Acheulean-like bifaces.

The Koshkurgan and Shoktas sites preserve the only known assemblages from a unique microlith industry [49]. Tools were primarily made from raw materials available in the vicinity, which were almost entirely small pebbles with an average size of 4 to 5 cm [54, 55]. Nearby Lower Paleolithic assemblages from the Karatau region such as Kyzyltau 1–2 [56], Borykazgan, Tanirkazgan, and Akkol [57] resemble a different technocomplex distinct from the small artifact industry of Koshkurgan and Shoktas. These assemblages contain general core-and-flake industries characterized by large chopping tools and proto-Levallois techniques [58, 59].

Pebble cultures are found in the Tajik depression and can also be considered a subset of core-and-flake industries [11]. Pebble technology is most prominent at Karatau but also identified at other nearby assemblages such as at Lakhuti/Obi-Mazar, Karamaidan, and Kul'dara. Reduction strategy is based on metamorphic pebble flaking, which produces a large portion of cortical flakes, fragments, and chips [60]. This industry is characteristic of the earliest dated material in Central Asia and is suggested to associate with the first migration wave out of Africa [47].

Finally, the desert plateaus and steppe regions of Kazakhstan, Uzbekistan, and Turkmenistan have yielded surface scatters of bifacially worked tools resembling the Acheulean. The clearest examples of these occur in the Krasnovodsk Peninsula on the Caspian Sea in Kazakhstan [11]. Surface scatters containing bifaces are uncommon but present throughout the Central Asian steppe. This includes eastern Kazakhstan in the Lake Balkash region (i.e., the sites of Semizbugu and Bale [61, 62]), and northeastern Kazakhstan (Kudaikol, Vishnevka-3 [63, 64]). It has been suggested that a second wave of Out-Of-Africa expansion brought an Acheulean bifacial industry by ca. 450–350 ka [47]. However, the chronology for this arrival remains uncertain because all Central Asian localities yielding bifaces lack stratigraphic context and sound chronologies.

In Central Asia, bifaces often occur alongside assemblages characteristic of Middle Paleolithic industries. For example, the Paleolithic sites at Kyzylnura ([65], reviewed in [11, 13]) are diagnostic of the Middle Paleolithic, but additionally contain large bifaces. The lack of secure dates and stratified contexts hinders strict assessment. It remains unknown whether these assemblages are truly representative of an Acheulean industry, a mix of multiple industries, a late developed Acheulean that occurred alongside Middle Paleolithic Industries, or perhaps even more recent bifacial traditions [66]. The absence of any evidence for dating associated with this typology makes it impossible to evaluate.

Vishnyatsky [11] reviewed and mapped the location of assemblages characterized by varieties of flake-and-core industries, and those containing bifaces. He noted that bifaces are known in the west and north, but are largely absent in the south, and generally associated with the plains. In contrast, core-and-flake industries occur almost exclusively in southern zones. At the time, Vishnyatsky [11] noted that one possible exception to this trend was Kulbulak, Uzbekistan, where handaxes and bifaces appear to be present in mountain foothills and at a

more southern latitude than expected [67]. However, according to Vishnyatsky [68] these bifaces are more likely Middle Paleolithic tools and more recently, Kolobova et al. [69] re-evaluated the Kulbulak assemblage and concluded that Acheulean tools are absent at Kulbulak altogether. Additional surveys have also recovered Acheulean-like bifaces in northern Kazakhstan at the Mugodzhar hills [70], in northeastern Kazakhstan at Ekibastuz 4 [71], and to the east of the Aral Sea [72, 73] with some of these possibly belonging to the Lower Paleolithic. These findings further support the notion that bifaces frequently occur in northern arid zones and are uncommon in mountainous, but hydrologically better supplied, areas.

## Taxonomy of early paleolithic toolmakers

Based on their antiquity, it is likely that many Central Asian Lower Paleolithic assemblages were formed by *Homo erectus*. Unfortunately, no hominin fossil remains are associated with Lower Paleolithic toolkits of Central Asia and fossils diagnostic of *Homo erectus* are absent altogether from Central Asia. Hominin remains recovered from Sel'Ungur were initially classified as *Homo erectus* [74]. This identification was later re-evaluated and the taxonomic attribution is now considered indeterminate [13, 75] and features of a juvenile humerus from Sel'Ungur are most similar to the morphology of Neanderthals or other archaic hominins [76]. The techno-typological features of the Sel'Ungur industry were also re-evaluated and are more consistent with the Middle Paleolithic [76]. Despite the lack of association in Central Asia between *Homo erectus* and toolkits, we contend that Lower Paleolithic assemblages were most likely formed by *Homo erectus*, because this taxon is the only known hominin in Asia during the time period when evidence of occupation first emerges (ca. 1.0–0.8 Ma) (Fig 2).

Hominin remains diagnostic of *Homo erectus* have been found at sites in neighboring regions, such as Kocabaş, Anatolia, Turkey [77], Nadaouiyeh An Askar, Syria [78], Gesher Benot Ya'aqov, Palestine [79, 80], Zhoukoudian [81] and Lantian Gongwangling [38], China, and on the island of Java, Indonesia [82, 83]. The estimated ages of these fossils place *Homo erectus* in Asia within and around the time frame that Lower Paleolithic toolkits are first discovered in Central Asia.

*Homo erectus* remains in Eurasia are linked to a variety of Lower Paleolithic tool types. Bifacial Acheulean traditions are associated with *Homo erectus* fossils from Nadaouiyeh An Askar, Syria [84, 85] and Gesher Benot Ya'aqov, Palestine [80]. However, the link between *Homo erectus* and Acheulean bifacial technology is complex [86] and bifaces are notably absent from several other assemblages associated with *Homo erectus*. Stone tools recovered from Zhoukoudian represent typical core-and-flake industries without bifaces [87–91]. Artifacts were not directly recovered at Kocabaş, Anatolia, although nearby lithics from Bozyer, Anatolia were suggested to be manufactured by *Homo erectus* [92]. These tools are also characteristic of a core-and-flake industry without bifaces [92]. At Sangiran, Java, Indonesia, *Homo erectus* remains were associated with non-lithic shell tools [93]. Due to the substantial technological diversity associated with *Homo erectus* in Eurasia, it is difficult to predict if certain technological features or technocomplexes in Central Asia are indicative of *Homo erectus*.

*Homo erectus* populations declined by the Middle Paleolithic and later industries overlap temporally with a variety of other hominin taxa. It is therefore unlikely that Middle and Late Paleolithic assemblages in Central Asia were produced by *Homo erectus*, and probable that these later traditions associate with some combination of Neanderthals, Denisovans, and modern humans. A lower premolar attributed to *Homo sapiens* has been recovered from Kulbulak, Uzbekistan [94]. Although no fossils of Paleolithic-age in Central Asia are securely diagnostic of Denisovans, this taxa is known nearby in Siberia during the Middle and Late Pleistocene [2, 95–98]. Neanderthal remains are also present in Siberia during the Middle and Late

Pleistocene [96, 97, 99–102], and have been uncovered in Uzbekistan at Teshik-Tash [103] and Obi-Rakhmat Grotto [104] alongside Middle Paleolithic assemblages. Hominin remains with morphological affinities to Neanderthals have also been uncovered at Sel'ungur [76], Anghilak Cave [6] and Khudji [105, 106].

**Objectives.** The presence of Lower Paleolithic assemblages in the steppe and deserts of arid Central Asia offers the possibility that these environments may once have formed habitable regions for early hominins, such as *Homo erectus*. In particular, the distribution of technocomplexes within the Lower Paleolithic, especially the geographic patterning of Acheulean bifaces, is intriguing and warrants a more systematic investigation. Here, we evaluate the importance of the steppe, semi-arid, and desert biomes to the earliest hominin occupation of Central Asia.

We consider archaeological evidence, with attention to the distribution of all finds, including undated surface scatters. This is because the majority of known Paleolithic sites come from surface occurrences, especially in the Middle Pleistocene. We define a "site" as any place where paleolithic stone tools were recovered in primary or sub-primary position. Changes in moisture availability are inferred from long-term climatic reconstructions and an annually resolved speleothem-based multiproxy record from the end of MIS 11 in Uzbekistan. We synthesize geographic trends in site distributions relative to the likely environments that existed when they were formed in order to interpret if and when the Central Asian plains were supportive of hominin habitation.

## Materials and methods

### Geographic distribution and elevation of Paleolithic assemblages

The geographic locations of 132 Upper Paleolithic, Middle Paleolithic and Lower Paleolithic assemblages were collected from the published literature (S1 Table). We broadly attributed sites to the Lower Paleolithic, Middle Paleolithic, and/or Upper Paleolithic, and noted the presence/absence of bifaces. We abstained from adopting a more specific taxonomic system for technocomplexes within these divisions because our goal was to investigate the spatial distribution of all sites, including surface scatters with limited context. Given the aforementioned limitations, it would be premature to divide many of these assemblages into finer-resolution categories. In cases where coordinates were not published, decimal degrees (dd) were approximated from maps. Elevation values were calculated in Google Earth Engine and using the ALOS 30 m global DEM. Because many sites were estimated from published maps and do not represent reported coordinates of a real location, we averaged the elevation of the zone around the location. Values were estimated for a 45 meter radius buffer. The latitudes and altitudes of Paleolithic sites were compared using Analysis of Variance (ANOVA) calculated in R Studio [107].

### A speleothem-based proxy record of hydrological changes

**Site and sample.** Amir Timur Cave is located in the arid westernmost reaches of the Zaravshan mountain ranges south of Samarkand in southern Uzbekistan (N39.422747°, E66.763206°). The regional climate is characterized by hot and arid summers and cold and humid winters. The cave is developed in marble at an altitude of 1813 m a.s.l. and exposed to western air masses. Most precipitation falls between autumn and spring [108]. The mean annual precipitation in the region ranges from 200–400 mm a$^{-1}$ [109]. Summer rainfall is rare and usually the result of local thunderstorms [108]. Backward trajectory modelling shows that moisture uptake in spring is mainly from the Middle East, but a shift from the Black Sea can be seen, with more moisture being taken up locally. Predominantly local moisture uptake from

the Caspian Sea and Aral Sea region is a characteristic feature of the summer months. From September to November, moisture uptake is more dispersed, with moisture sources ranging from the Mediterranean, across Central Europe to the Middle East. In winter, moisture is mainly supplied from the Caspian Sea and Aral Sea regions. Throughout the year, the Mediterranean is a source of moisture. Moisture flux from the north and northwest dominate southeast Uzbekistan while no moisture is delivered from south or southeast.

Vegetation cover is minimal above the cave, with only grasses and a few shrubs near the entrance and a thin soil cover of <15 cm. Amir Timur Cave is very dry with only a few areas with dripping water. The stalagmite discussed here (S-12-4) was found as a broken fragment with a length of 76 mm. Petrographic and mineralogical observations indicate that stalagmite S-12-4 is characterized by light, porous and dark, dense calcite layers (S1 File, S1–S3 Figs).

**Uranium-thorium dating.** Six powder samples were taken from S-12-4 with an ESI New Wave micromill at the Max Planck Institute for the Science of Human History, Jena and dated using the $^{230}$Th/$^{234}$U dating method at the Institute of Global Environmental Change, Xi'an Jiaotong University. The samples for U/Th dating were drilled parallel to the horizontal course of the layering (S4 Fig). After surface cleaning, approximately 300 mg of material was collected for each sample. Chemical separation of Uranium and Thorium was carried out similarly to the procedures outlined in Edwards et al. [110]. The concentrations of uranium and thorium isotopes were analyzed via multi-collector inductively coupled plasma mass spectrometry (MC-ICP-MS, Thermo-Scientific Neptune), and the instrumental approaches are described in detail in Cheng et al. [111]. A $^{230}$Th/ $^{232}$Th atomic ratio of $4.4 \pm 2.2 \times 10^{-6}$ was used to correct for the initial $^{230}$Th. Since all the measured $^{230}$Th/ $^{232}$Th atomic ratios are larger than $900000 \times 10^{-6}$, the corrections of initial $^{230}$Th for all samples are negligible. The measured isotope ratios of uranium and thorium, the decay constants and the calculated ages are listed in S2 Table.

**Greyscale values and layer counting.** Greyscale values were gained using the open-source software ImageJ, version 1.53k (https://imagej.nih.gov/ij/download.html) on a high-resolution scan (3200 dpi) of the stalagmite. Greyscale values were extracted between the first clearly visible bright layer and U-Th sample U8 (S4 Fig) along the growth axis (from 3.5 mm to 62.0 mm from top) and recorded with a resolution of ca. 7.9 μm. Individual growth layers were counted based on the greyscale record (S5 Fig). Grey value peaks associated with brighter layers were manually counted over the entire greyscale profile. A total of 780 layers were identified, averaging 71.4 μm per layer. The distance between each identified peak is taken as annual carbonate deposition assuming that each layer represents a whole year.

**Stable carbon and oxygen isotopes.** Stable carbon and oxygen isotopes ($\delta^{13}$C and $\delta^{18}$O) were measured at Northumbria University. The samples were obtained at the Max Planck Institute for the Science of Human History, Jena, following the procedure of Baldini et al. [112], using an ESI New Wave micromill. Samples were milled continuously at ca. 210 μm intervals. A total of 268 high resolution samples and two starting samples with lower resolution were measured on a ThermoScientific Delta V Isotope Ratio Mass Spectrometer (IRMS) coupled with a ConFlo IV and a Gasbench II following the procedure of Spötl and Vennemann [113]. The results are presented in per mil against Vienna PeeDee Belemnite (VPDB). The external standard deviation is <0.1 ‰ for $\delta^{13}$C and $\delta^{18}$O.

**Laser ablation ICP-MS analysis.** Laser ablation (LA-) Inductively Coupled Plasma Mass Spectrometry (ICP-MS) was performed at the Centre for Microscopy, Characterisation and Analysis at the University of Western Australia. A Teledyne Analyte G2 UV-excimer laser (193 nm) was coupled to a Thermo-Fisher Element XR High Resolution (HR-)ICP-MS, using a squid signal smoothing device. An effectively continuous laser ablation raster was performed along the growth axis of speleothem comprised of 21 shorter rasters, allowing interspersion of

primary and secondary reference materials. A spot size of 180 by 30 μm was used at a laser repetition rate of 10 Hz and a raster rate of 4 μm/s. A full range of analytes were included with a cycle time ca. 1.3 seconds, however only Mg/Ca is reported in this study using $^{25}$Mg and $^{43}$Ca as the isotopes of interest. Data reduction was performed in Iolite, including correction of detector analogue-counting factor, instrumental baselines, and standard sample bracketing. NIST 614 was used as the primary reference material, with NIST 612 and a range of in-house carbonates (low-Mg calcite, high-Mg calcite and aragonite) used as secondary reference materials. These secondary reference materials indicate Mg/Ca reproducibility better than 2.2% RSE on all standards (n = 11, n = 5, n = 10, n = 10, n = 10). Four 2 mm long Mg/Ca rasters performed along individual laminations show comparable reproducibility.

## Results

### Geographic distribution of assemblages

The locations of Lower Paleolithic, Middle Paleolithic, and Upper Paleolithic finds do not significantly differ in latitude or altitude according to industry (Fig 3). Estimated meters above sea level (m a.s.l.) were similar (ANOVA: $F_{(2,179)}$ = 2.01, p = 0.137) across Lower Paleolithic (Mean = 599.1, SD = 604.0), Middle Paleolithic (Mean = 756.1 m a.s.l., SD = 615.3), and Upper Paleolithic (Mean = 869.6 m a.s.l., SD = 792.8) site complexes. This was also the case for latitude (ANOVA: ($F_{(2,179)}$ = 2.11, p = 0.124) across the Lower Paleolithic (Mean = 44.8 decimal degrees (dd), SD = 4.1) Middle Paleolithic (Mean = 43.3 dd, SD = 4.1) and Upper Paleolithic (Mean = 44.0 dd, SD = 4.3).

Despite overall similarities across Paleolithic subdivisions, within the Lower Paleolithic, differences in spatial distribution were detectable in assemblages that included bifaces (Fig 4). Bifaces are associated with significantly higher latitudes (mean = 45.0 dd, SD = 3.9) relative to all other Paleolithic assemblages (mean = 42.8 dd, SD = 4.2; ANOVA: $F_{(1,130)}$ = 13.08, p < 0.001). Bifaces were also recovered at significantly lower altitudes (mean = 345.8 m a.s.l., SD = 279.5) compared to other Paleolithic assemblages (mean = 965.0, m a.s.l. = 762.2; $F_{(1,130)}$ = 16.53, p < 0.001).

### Speleothem age model–combining U/Th dating and layer counting

The U/Th age model based on 6 U/Th dates from S-12-4 shows high uncertainties of several thousand years (S6 Fig). The stalagmite was deposited between 405±8 ka BP and 387.8±7 ka BP. Due to large U/Th dating uncertainties we constructed an additional age model based on layer counting information from the greyscale profile, assuming that the identified layers represent annual growth. The layer counting revealed a growth period of ca. 780 years. The layer counting record was related to the U/Th chronology by linking the layer count record to the median U/Th age of 391202 years. While this strategy links the layer counting chronology to the U/Th age model, it leaves the former floating within the large radiometric errors of several thousand years of the latter.

To constrain the median and estimate the 95% confidence range of the proxy time series 2000 Monte Carlo based age-depth simulations were calculated using the COPRA routine [115]. The resulting layer counting chronology has a much smaller internal error compared to the U/Th dates allowing more detailed insights into inter-annual growth rate and hydrological changes and the underlying environmental processes even if the absolute temporal framework remains relatively poorly constrained.

### Speleothem greyscale values

Greyscale values vary between 53 and 151, with a median value of 96. High greyscale values represent brighter porous calcite layers, while lower greyscale values reflect dark, dense layers

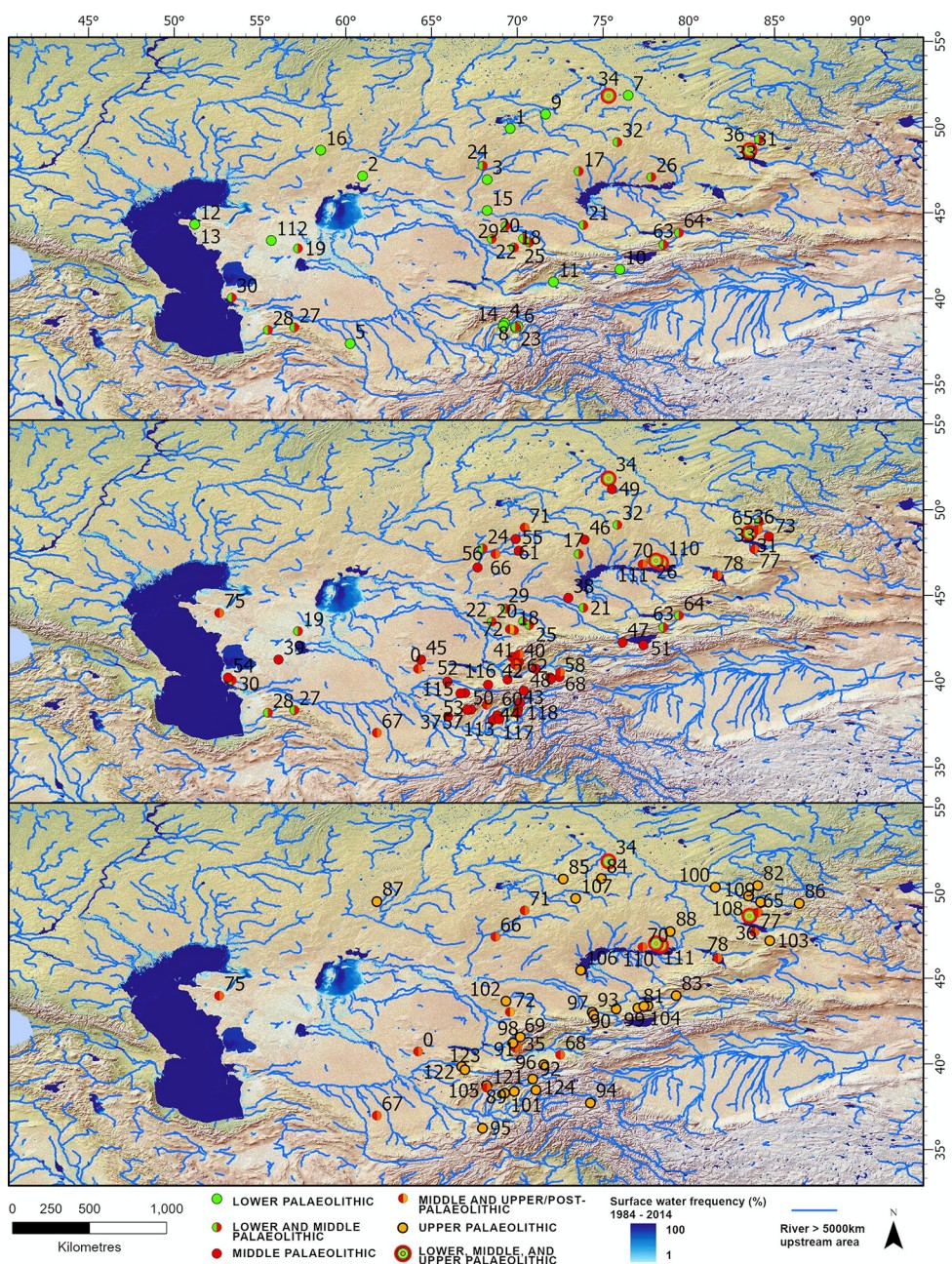

**Fig 3.** Distribution of Lower Paleolithic (top), Middle Paleolithic (middle) and Upper Paleolithic (bottom) finds. Assemblages that contain multiple paleolithic industries are color-coded accordingly (green = LP, red = MP, orange = UP) and displayed on all applicable maps. Numbers correspond to the sites listed in S1 Table. Surface water [15], and rivers (Hydrosheds [16]) are displayed. Basemap is Natural Earth 2 data.

(see Fig 5A). The 780 counted layers show an average growth rate of 71.4 μm, a typical value for speleothems [112]. The minimum growth rate is 15.8 μm and the maximum 277.8 μm.

Greyscale values increase from ~391.67 to 391.61 ka BP from around 98 to 148. After that they show a general decreasing trend to lower values until ~391.4 ka BP. During this ~210 year long continuous decline we observe three pronounced decreases at ~391.58, 391.54 and 391.51 ka BP. These shifts to lower values are separated by higher values with a maximum at ~391.55

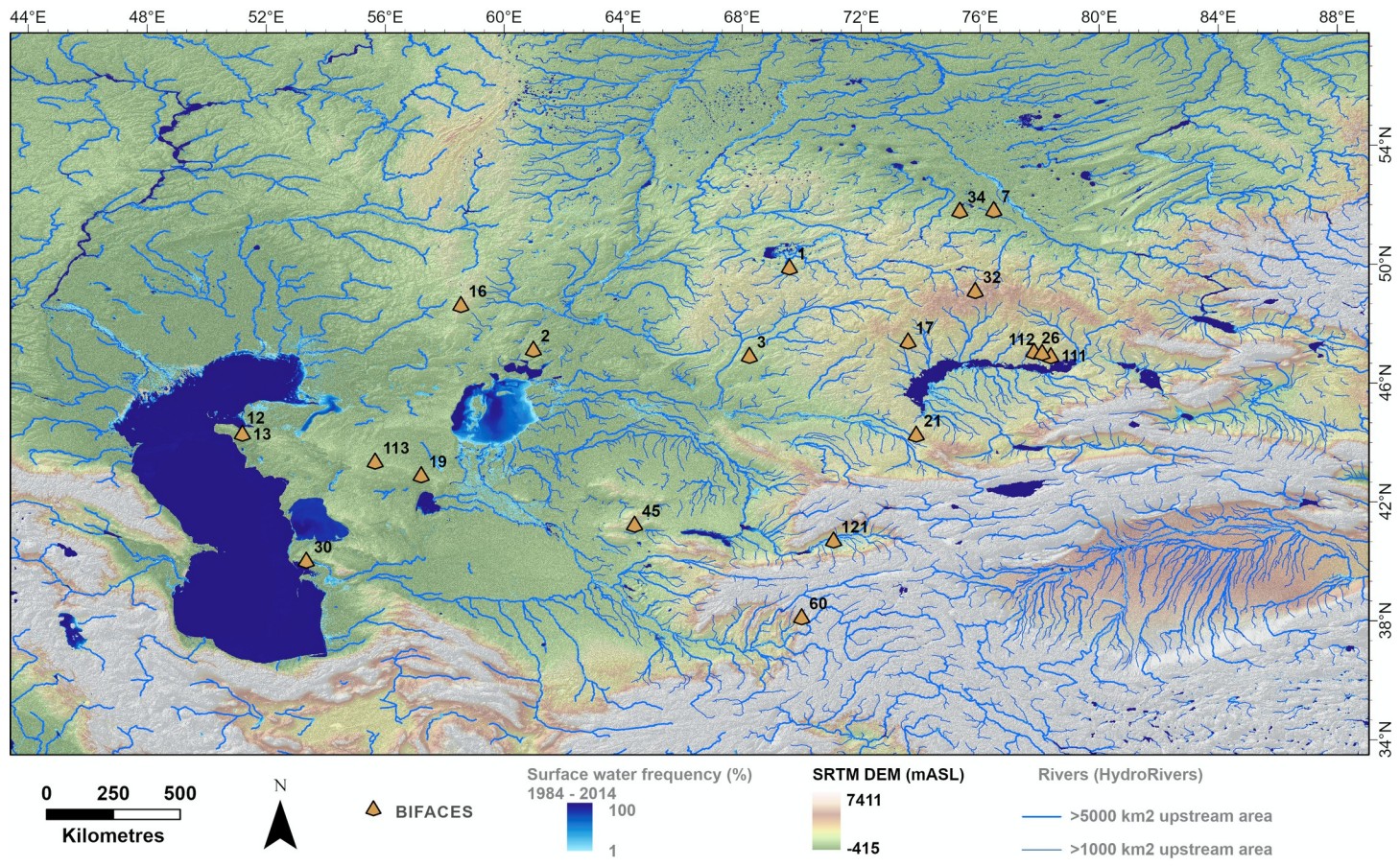

**Fig 4. Distribution of acheulean-like bifaces.** Elevation data for Central Asia (SRTMV4 [114]) with bodies of water [15], rivers (Hydrosheds [16]) and the boundaries of Central Asia outlined. Areas where bifaces have been recovered are marked (orange triangle). Numbers correspond to the sites listed in S1 Table.

ka BP. Notably, the shifts to lower values around 391.54 and 391.51 ka BP are mirrored by higher values in the $\delta^{13}$C record. The decrease in greyscale values is followed by three shifts to higher values (up to ca. 116) between ~391.4 and 391.32 ka BP. These shifts co-occur with similar trends in the stable isotope and Mg/Ca records. A distinct decrease in greyscale values occurs at ~391.32 ka BP, followed by a rapid increase (about 146) at ~391.27 ka BP. Afterwards, greyscale values fluctuate around 125 for ~10 years and rise again at ~391.26 ka BP. A fast (~30 years) decline occurs between ~391.26 and 391.23 ka BP. The rapid change between ~391.32 to 391.23 ka BP is reflected in all proxies. Between ~391.23 and 390.97 ka BP the greyscale profile shows an arc-shaped progression towards higher values up to ~133. During this period several prominent shifts towards higher values occurred. These correlate with a drop in Mg/Ca and stable isotopes. After ~390.97 ka BP greyscale values rise from about 65 to around 126 within ~70 years. Over the entire 780 record from ~391.68 to 390.9 ka BP we observe a trend towards lower greyscale values (see Fig 5B).

Both Mg/Ca and stable isotopes show similar dynamics. High Mg/Ca and stable isotope values change to very low values within a short period of time (about 50 years), see below.

Fig 5A shows a clear correlation between greyscale values and layer coloration. The relationship between layer coloration and the individual peaks of the graphs can be seen particularly in the horizontal layer track, on the right side of the stalagmite. Dark layers are represented by lower greyscale values and higher greyscale values reflect brighter layers. This is

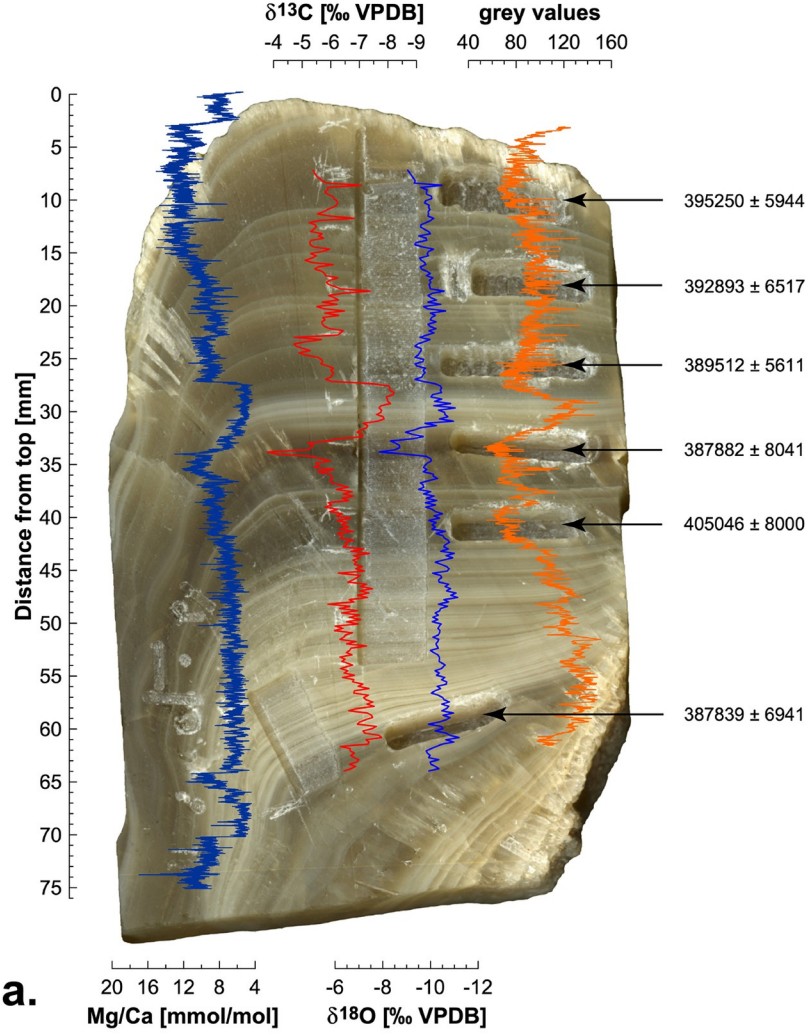

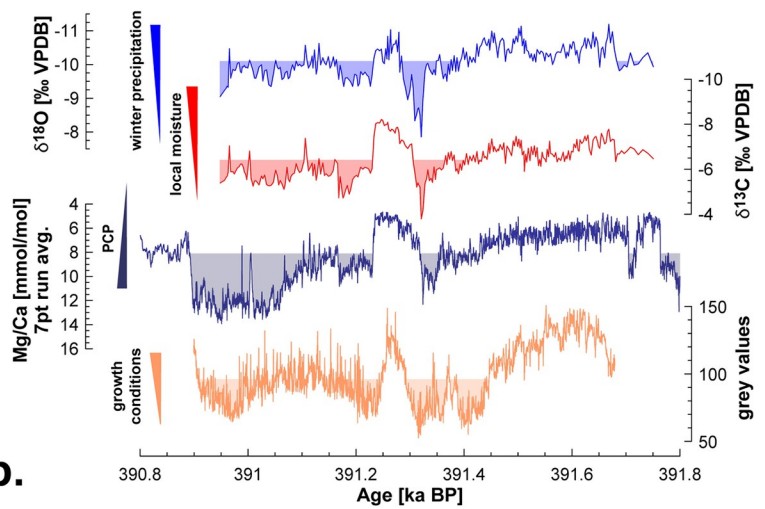

**Fig 5. Time series of Mg/Ca, δ¹³C and δ¹⁸O proxies for stalagmite S-12-4.** A: An overview of stalagmite S-12-4 with Mg/Ca, δ¹³C, δ¹⁸O, and greyscale value profiles. Mg/Ca, δ¹³C and δ¹⁸O correlate positively, while greyscale values partly co-vary, albeit with increased individuality, especially in the lower half of the stalagmite. B: Final proxy time series of S-12-4, based on layer counting linked to the median U/Th age. Shaded fillings are cut off at the median value of the respective proxy.

especially evident in the range between around 32 and 34 mm from top. The dark area is characterized by lower greyscale values until minimum, in the overlying area between ~32 and 28 mm there are noticeably higher greyscale values, which are related to the distinct light layers of the stalagmite.

## Stable isotopes

The δ¹³C values vary between -8.21 and -3.79 ‰, with a median of -6.39 ‰, while δ¹⁸O range between -11.2 and -7.86 ‰, with a median of -10.1 ‰. There is a significant positive correlation between δ¹³C and δ¹⁸O across the profile (r = 0.71, $p<0.001$, n = 270). Like for the greyscale record, strong fluctuations are also observed in the isotope profiles (Fig 5B).

Starting from ~391.75 to 391.69 ka BP, δ¹⁸O values fluctuate between around -10.4 and -9.8 ‰. Thereafter, the values start to decrease until ~391.67 ka BP to the minimum. From about 391.67 ka BP, δ¹⁸O values generally increase to ~391.53 ka BP. During the rise, two more conspicuous shifts (~391.65 and 391.61 ka BP) towards higher values occurred, well reflected in δ¹³C and Mg/Ca. Starting around 391.53 ka BP, δ¹⁸O values decrease again to circa -11.2 ‰ by ~391.51 ka BP. Subsequently, values rise to ~391.36 ka BP to about -9.6 ‰. During this period two noticeable shifts to higher values at ~391.48 and 391.46 ka BP can be discerned. These shifts are also found in δ¹³C and Mg/Ca. In the same period, the greyscale values decrease. Thereafter, the δ¹⁸O shift to lower values (ca. -10.3 ‰), centered around ~391.35 ka BP.

Over the discussed period δ¹³C behaves very similar to δ¹⁸O. Between ~391.75 and 391.68 ka BP, values fluctuate between ca. -7 and -6.3 ‰. From ~391.68 ka BP, δ¹³C shows a short drop to around -7.8 ‰. From then on values generally rise until ~391.34 ka BP to ca. -5.4 ‰. During the discussed period, five more pronounced shifts to higher values are observed at around 391.63, 391.54, 391.51, 391.48 and 391.39 ka BP, with some correlating with higher δ¹⁸O and Mg/Ca ratios and lower greyscale values. Several small shifts towards lower values, concentrated around 391.56, 391.53, 391.5, 391.45 and 391.37 ka BP, are also reflected in the other proxies (see Fig 5B).

Both stable isotope records drop slightly by ~391.33 ka BP and rise afterwards rapidly until ~391.32 ka BP to their maximum values. Subsequently, δ¹³C declines to its minimum at ~391.25 ka BP while δ¹⁸O declines rapidly to about -11‰ (ca. 391.28 ka BP). From then on, δ¹⁸O values fluctuate dynamically between ca. -11 and -10 ‰ to ~391.24 ka BP. Starting at ca. 391.24 ka BP, both isotopes increase rapidly within ~10 years. δ¹⁸O increases from around -10.5 to -9.3 ‰ and δ¹³C increases from about -8.0 to -5.8 ‰. This strong increase occurs parallel to a fast increase in Mg/Ca. Starting at ~391.23 ka BP, both isotope ratios drop slightly and then rise slowly until ~391.17 ka BP, with a stronger increase in δ¹³C. This shift is also mirrored in the Mg/Ca ratio at that time. Between around 391.17 and 390.95 ka BP, δ¹³C fluctuates dynamically between around -7.4 and -4.7 ‰. During this period three prominent peaks with lower δ¹³C values occur at around 391.11, 391 and 390.97 ka BP. These minima are reflected in the δ¹⁸O and Mg/Ca ratios. The δ¹⁸O values vary also strongly from ~391.17 to 390.95 ka BP between about -10.6 and -9 ‰. Especially, the striking shifts in the stable isotopes are concurrent with shifts in the Mg/Ca ratio (see Fig 5B).

Both isotope profiles (Fig 5A) also show a correlation with the layer coloration of stalagmite S-12-4. This is particularly evident in the sampled area. Lower isotope values concur with light

layers, while darker layers occur with higher isotope values. This is well expressed in a strong change of values between ~28 and 34 mm from the top. In the darkest section of the stalagmite, approximately between ~32 to 34 mm from top, the isotope values reach their maxima. In the very light area above, at about 28 to 32 mm, $\delta^{13}C$ reaches its minimum while the $\delta^{18}O$ values also decline strongly.

## Laser ablation Mg/Ca element data

Fig 5B shows the Mg/Ca ratio with a moving average in an interval of 7 points. The Mg/Ca ratio indicates a minimum of 4.66 mmol/mol and a maximum of 13.94 mmol/mol, with a median of 8.1 mmol/mol. Like carbon, oxygen isotopes and greyscale values, Mg/Ca ratio fluctuates notably over the discussed time.

Mg/Ca decreases in general from about 13 to around 4.7 mmol/mol between ~391.8 and 391.74 ka BP, after which values rise rapidly to around 10.2 mmol/mol until ~391.71 ka BP. Immediately afterwards, the Mg/Ca values drop sharply to around 5 mmol/mol within ~10 years. From ~391.7 to 391.38 ka BP a continuous increase in the Mg/Ca is evident, with values rising from about 5 to 9.8 mmol/mol in ~320 years. During this period five more prominent shifts to higher values are found at around 391.65, 391.62, 391.61, 391.54 and 391.51 ka BP. Several of these peaks are reflected in the stable isotope ratios and correspond with lower greyscale values. Individual small shifts to lower values, for example at ~ 391.67 and 391.5 ka BP, are reflected in the $\delta^{13}C$ and $\delta^{18}O$ ratios. Starting at ~391.38 ka BP onwards, two successive shifts to lower values at around 391.37 and 391.36 ka BP are observed. Whereas the first one is best seen in the $\delta^{13}C$ and greyscale records. From ~391.36 ka BP, the Mg/Ca values rise to around 11.4 mmol/mol within ~20 years. Thereafter, the values drop for a short time and then rise rapidly until ~391.32 ka BP to about 12.3 mmol/mol. After this increase, the values sink to their minimum at ~391.25 ka BP and then rise very quickly within around 20 years, to about 10.1 mmol/mol. These rapid changes co-occur with the other proxies mentioned above (see Fig 5B). From ~391.23 ka BP, the Mg/Ca values initially decrease slightly, together with the stable isotopes, then rise to around 10.8 mmol/mol by ~391.17 ka BP. The increase is also especially well visible in $\delta^{13}C$ record. Up to ~391.16 ka BP Mg/Ca values decrease and then rise to about 13.5 mmol/mol until ~391.03 ka BP. During this period several single shifts to lower values occur, e.g., centered around 391.11, 391.07, and 391.05 ka BP. Most peaks are also found in the stable isotope records and with higher values in the greyscale record.

Starting at ~391.03 ka BP, Mg/Ca shows a trend to lower values until ~390.95 ka BP. Two strong shifts up to lower values to about 7.5 mmol/mol are prominent at ~391 and 390.99 ka BP. Both peaks are also seen in $\delta^{13}C$, whereby the first is more pronounced. At the same time the greyscale values increase. At ~390.95 ka BP, the Mg/Ca values increase to their maximum. From ~390.95 to 390.9 ka BP, Mg/Ca values decrease at first slowly and from ~390.9 ka drop rapidly within ~10 years to around 6.2 mmol/mol. Mg/Ca values rise again until ~390.87 ka BP and fluctuate from there between ~7.3 and 9.2 mmol/mol, but fall from ~390.82 to 390.8 ka BP to about 6.6 mmol/mol. The Mg/Ca ratio shows a general increasing trend from ~391.8 to 390.8 ka BP.

Fig 5A shows the original Mg/Ca record and displays no strict correlation with the grey values although lower Mg/Ca values coincide with the brighter interval between ~28 and 32 mm from top. The minimum is located shortly above the bright layered interval. In the underlying dark section (~32 and 34 mm from top) the Mg/Ca values strongly increase.

Mg/Ca and stable isotope ratios, as well as the greyscale values show a strong change from higher and lower values over ~90 years between ~391.32 and 391.23 ka BP. In particular the $\delta^{13}C$ and Mg/Ca time series show a similar pattern, especially with the coincidence of the individual peaks.

## Discussion

Lower Paleolithic artifacts are scattered throughout arid Central Asia, from the Karakum and Kyzylkum deserts of Turkmenistan and Tajikistan, to the desert plateau between the Caspian and Aral Sea, and Lake Balkash and the Zaisan paleo-lake basin in Kazakhstan. Consistent with previous observations [11], we find that bifaces are disproportionately concentrated at higher latitudes and are prevalent through the low and mid-altitude plains of Central Asia (Fig 4). In contrast, hominins forming other Paleolithic assemblages are found more frequently in the mountain foothills. The association of the northern plains with bifaces and southern Central Asia with other paleolithic industries could reflect differences in local raw materials availability, tool function, the timing of habitation, or the groups of hominins forming the assemblages. It is difficult to assess which combination of factors explain this observation because evidence for bifacial industries in Central Asia is extremely limited. Below we consider how climatic variables, the preferences of hominin taxa forming assemblages, and/or preservation bias could explain the observed distribution of Paleolithic assemblages and the occupation of arid Central Asia.

### Climatic factors

**Long-term low-resolution climatic trends.** During the time interval when Lower Paleolithic assemblages were formed, super-interglacials MIS 15–11 could have provided temperate conditions and water supplies that allowed the periodic occupation of higher latitudes. The climate of arid Central Asia is closely tied to the westerlies, which transport moisture to this region [17]. The loess record from Tajikistan (Fig 2) shows that before the MPT changes in aridity and humidity were muted, but coincident with glacial interglacial cycles. However, since the MPT the region has experienced increased fluctuations in moisture availability [26]. These changes occurred primarily because the strength of the westerlies is influenced by glacial-interglacial sea surface temperatures via their control on evaporation from the Atlantic Ocean. As westerly weather systems move eastwards from the Atlantic Ocean, they are enhanced by evaporation of the waters from the Mediterranean, Black, Aral and Caspian Sea [21], and this would have increased during interglacial warm periods [116, 117]. Furthermore, when the Black and Caspian Sea were larger, additional evaporation from the increased surface area of water would have enhanced the moisture budget of the westerlies even further. Given this, the interglacial moisture supply would likely be highest when the Black and Caspian Seas experienced high stands that were coincident with high North Atlantic SST, allowing increased evaporation, with westerlies transporting this moisture directly into Central Asia. Such conditions likely occurred when Caspian Sea high stands coincided with long interglacial periods, and deglaciation periods.

This mechanism for transporting moisture into Central Asia can potentially explain the occupation of areas that are now desert at times when they were more humid. Firstly, the Caspian Sea transitioned to a long high stand between 880–750 ka (Fig 6) [19]. This is also the period when the earliest archaeological localities emerge. Between approximately 600 to 500 ka (during MISs 15–13), the loess record (Fig 2) suggests that this was a particularly significant humid period in Asia, with long interglacials during MIS 15 and 13 and only limited northern ice extent during the MIS 14 glacial [118]. Furthermore, it was a period when there were prolonged high stands in the Caspian and Black Seas (Fig 2). This long period of high humidity has previously been suggested to have facilitated hominin dispersals into Asia [118] and may have contributed to the appearance of Acheulean-like bifaces in China during this time interval. In Central Asia, many Lower Paleolithic assemblages coincided with both high Caspian Sea levels and extended MIS 15–13 humidity, and they overlap geographically with what are

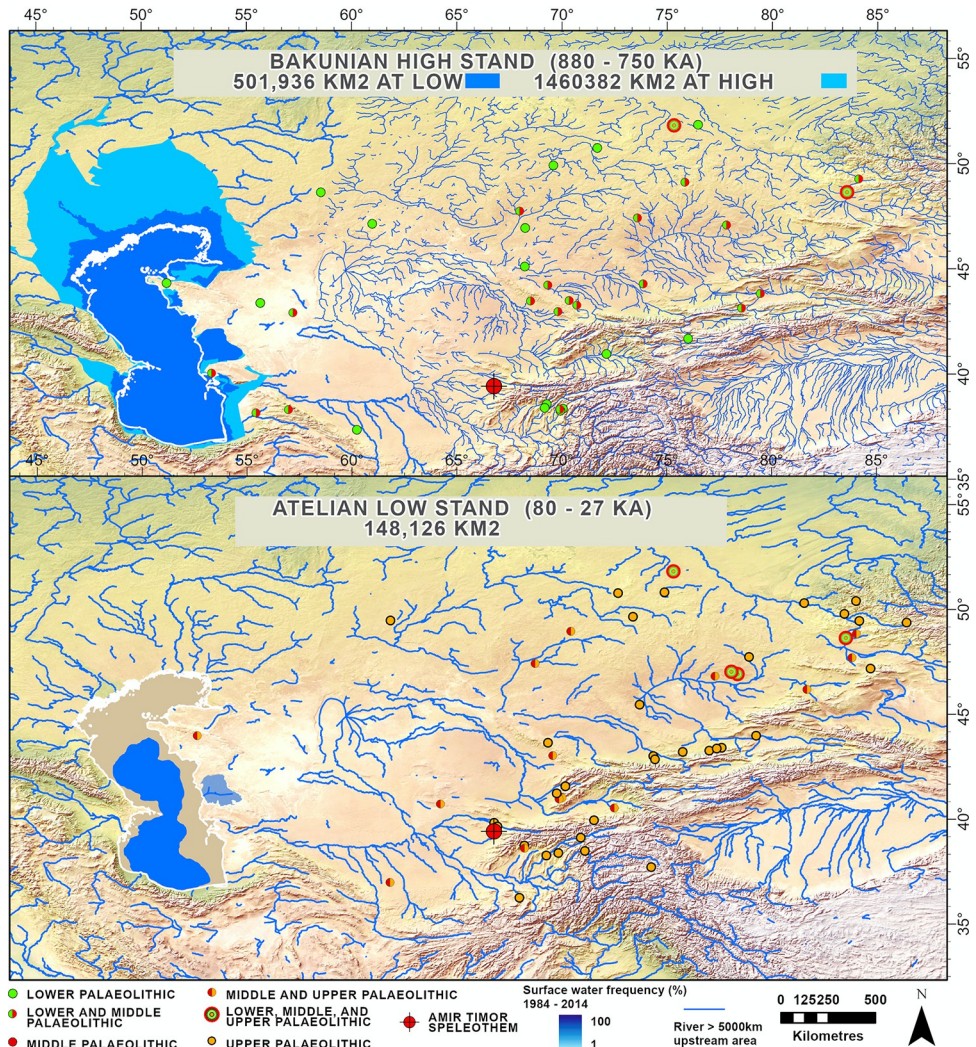

**Fig 6. Models of the Bakunian high stand and Atelian low stand water supply calculated by the authors, with paleolithic finds (per [19]).** Lower Paleolithic finds are plotted on the map of the Bakunian high stand and Upper Paleolithic finds are plotted on the map of the Atelian low stand. The location of the Amir Timur speleothem is also plotted (red target). Basemap is Natural Earth 2 data.

currently typically arid zones. The Koshkurgan faunal complex in Kazakhstan also supports a relatively warm and rather humid climate during the first half of the Middle Pleistocene with the composition of fossil animals indicating the existence of both forest and steppe landscapes [12, 119]. Taken together, this suggests that occupation during this time interval could have been sustained by more abundant water supplies and more temperate/humid conditions. Interestingly, it has been suggested that the same long warm period also set the stage for a period of extensive dispersal of hominins into Europe [120].

Against the backdrop of increasing aridity in Central Asia through the Middle Pleistocene, the region became progressively less habitable. The composition of the Late-Middle Pleistocene fauna testifies to cooling and the dominance of open steppe spaces overgrown with grasses and bushes [12]. It is likely that as conditions became drier and more extreme, hominins took refuge in the mountain foothills and river galley oases, which would have provided a more habitable environment than steppe and desert zones during both glacial and interglacial

periods [4]. Indeed, the spatial distribution of Middle and Upper Paleolithic sites supports this view, with both showing a variety of sites in the mountain foothills and few in the desert (Fig 3). Ecological models suggest that the mountain foothills may have served as refugia for hominins through the Late Pleistocene, offering more predictable water supply and a more attractive landscape than the adjacent open steppe (i.e., temperatures and seasonality are less extreme, and conditions are wetter during both glacials and interglacials) [4].

**A regional high-resolution multi-proxy record from S-12-4.** While multi-centennially to decadally resolved proxy records provide valuable insights in long-term (glacial-interglacial) developments of regional climate, they can only give limited information on environmental variability at the timescales relevant for hominin decision making. Using multiple paleoenvironmental proxies within layer counted chronologies, we can gain glimpses of hydrological changes on seasonal to multi-decadal scales. This level of information allows a significantly improved interpretation of human activities reflected in the archaeological record. Hydrological changes on seasonal to decadal timescales certainly would have been a major factor influencing humans migrating into and through the Central Asian landscape. Decisions on movement through a landscape were made at timescales ranging from days to years, without regard to multi-centennial developments.

The stalagmite from Amir Timur Cave reveals a ca. 780-year-long period of short-term hydrological changes in southern Uzbekistan at the end of the Marine Isotope Stage 11. Regional climate was by no means stable at that time or characterized by a simple trend towards colder and/drier as low-resolution records might suggest [26, 121] (Fig 2). The (longer-term) co-variation of the grey value, $\delta^{13}C$, $\delta^{18}O$, and Mg/Ca records (Fig 5) suggests a common environmental forcing.

Variations in grey values from a scanned stalagmite surface reflect changing crystal fabrics, with dense microcrystalline calcite being more translucent and thus associated with lower grey values. Faster growing calcite is frequently characterized by elongated crystal columns [122] with high interstitial space and many crystal boundaries, resulting in higher grey values. Speleothem growth rates are influenced by water and dissolved inorganic carbon supply as well as $CO_2$ degassing. Enhanced infiltration would result in faster growth, which would be reflected in lighter crystal layers with higher grey values. Reduced growth results in darker and thinner layers.

The $\delta^{13}C$ record is influenced by several mechanisms, including vegetation composition and activity, soil microbial activity, prior carbonate precipitation (PCP) [123] within the epikarst above the cave, and kinetic fractionation occurring when $CO_2$ degasses from dripwater entering the cave [112, 124]. Reduced effective precipitation lowers vegetation and microbial activity, allows increased epikarst PCP, and (eventually) lowers drip rates in the cave, which in turn intensifies isotopic fractionation through prolonged $CO_2$ degassing. All these factors ultimately result in higher $\delta^{13}C$ values in the stalagmite during drier conditions, thus making $\delta^{13}C$ a valuable tracer of *local* hydrology at seasonal to multi-annual timescales. Increased soil $CO_2$ exchange under open system conditions in the epikarst can counteract the factors mentioned above and act to lower speleothem $\delta^{13}C$ values [125]. However, this mechanism is often outpaced by the kinetic processes.

The stalagmite $\delta^{18}O$ signal potentially reflects several processes that include i) moisture source dynamics, ii) amount of precipitation, iii) temperature and iv) seasonality of precipitation [e.g., 126–128]. Additionally, mineralogical changes between calcite and aragonite could affect speleothem $\delta^{18}O$ values. In the present case, moisture is delivered to Amir Timur Cave mostly during the winter season as mentioned above [17, 129]. In the hot summers, only localized thunderstorms sporadically deliver rainfall. Infiltration and epikarst recharge are thus strongly biased towards winter (see also figure S3 in [30]). No moisture is derived from the

Indian summer monsoon realm. An amount effect (i.e., a negative correlation between rainfall $\delta^{18}O$ and amount of precipitation) is certainly at play at event scale (i.e., when raindrops are partly re-evaporated during their fall through dry air during summer thunderstorms) but given the insignificant contribution of summer rainfall to the total effective infiltration, this effect is outcompeted by seasonal scale precipitation and temperature changes. Rainfall $\delta^{18}O$ in Central Asia shows a significant correlation with seasonal temperatures, with higher values occurring during the hot summer [30]. However, given the minimal contribution of summer rainfall relative to winter precipitation this temperature signal is unlikely to be recorded in the stalagmite. Detailed Raman mapping along the growth axis of the stalagmite (S1 File, S1–S3 Figs) shows that S-12-4 is entirely composed of pristine calcite and mineralogical effects on $\delta^{18}O$ can be excluded.

Considering the above lines of evidence, we interpret the S-12-4 $\delta^{18}O$ signal as indicative of the relative contribution of winter vs. summer precipitation. Increased winter precipitation results in lower $\delta^{18}O$ values in the speleothem and (relatively) increased warm season precipitation would be reflected in higher $\delta^{18}O$ values in the stalagmite. Given the limited overburden, it is possible that (summertime) evaporation of soil water and PCP could amplify this pattern further.

The magnesium (Mg) concentration and thus the Mg/Ca ratio in calcitic stalagmites frequently indicates PCP dynamics. Under drier conditions, Mg is retained in the infiltrating water while calcium is preferentially removed if carbonate precipitates under subaerial conditions in the epikarst [123].

In stalagmite S-12-4 we observe covariation between $\delta^{13}C$, $\delta^{18}O$, and Mg/Ca (Fig 5), which can be explained by the interplay between changes in winter moisture supply and increasing kinetic fractionation at times when infiltration decreases (dry season length and intensity) as outlined above. Reduced winter precipitation relative to the annual budget is reflected in higher $\delta^{18}O$ values. At the same time, longer and/or drier summers increase PCP and reduce drip and growth rates, which would lead to lower greyscale values, higher $\delta^{13}C$ and Mg/Ca ratios respectively. Long-term processes like host rock dissolution dynamics or centennial-scale drying trends can additionally affect the Mg/Ca recorded in the stalagmite. In fact, we observe increasing Mg/Ca ratios over the entire stalagmite record, which we interpret as long-term drying trend with the onset of the MIS10 glacial.

The S-12-4 record reveals three multi-decadal scale drying episodes, with the oldest one centered around ca. 391.32, the second at ca. 391.23, and the last at ca. 390.95 ka BP (Fig 5). The drying events lasted each for about 40 years, separated by somewhat wetter conditions. Noticeably, drying began gradually but wetter conditions set in relatively rapidly. The first dry interval was also the most severe and followed by the wettest interval. This wet phase lasted for ca. 70 years, centered around 391.27 ka BP. The change from the driest to the wettest conditions took place within about 50 years. The Mg/Ca profile follows a similar pattern, implying that drying affected the epikarst. It is very likely that drying was initiated by a decline of winter precipitation which would lead to insufficient recharge of the epikarst, and thus to PCP. This lack of infiltration would reduce drip rates and allow for longer $CO_2$ degassing, reflected in higher $\delta^{13}C$ values [124].

The observed multidecadal to centennial changes in local hydrology might hint at periodic variations in winter moisture supply and atmospheric circulation patterns linked to the North Atlantic. Significant periodicities between 50 and 130 years of the Atlantic Multidecadal Oscillation [130] could potentially be linked to changes in Central Asian winter precipitation, with warmer conditions in the North Atlantic being more conducive for moisture uptake and transport towards Uzbekistan. While physically possible, this tentative link between the North Atlantic and Central Asia requires further in-depth evaluation.

Stalagmite S-12-4 exemplifies how highly resolved multi-proxy records can contribute to decipher local and regional environmental dynamics and potentially the physical forcings behind these changes. Periodic warmer and wetter phases could have supported hominin occupation in more northern arid regions. Hydrological variations on decadal timescales suggest that significant intervals of wetter conditions spanned substantial periods of time even during overall climatic downturns towards colder and drier conditions. Because favorable wet conditions appeared to develop quickly relative to the onset of drying events, hominins could have taken advantage of adjacent plains periodically and retreated when necessary. Within super-interglacials, like MIS 11, multi-decadal wetter periods would have supported dispersal and occupation of currently arid regions. It has been suggested that many assemblages with bifaces were likely formed sometime between 450–350 ka [47] and ESR dates indicate that hominins occupied arid regions near the Kyzlkum desert at Koshkurgan and Shoktas during MIS 13 and/or MIS 11 [49, 131]. These occupations were perhaps supported by regional multi-decal favorable periods against the backdrop of long-term glacial-interglacial trends.

## Hominin habitat preference

The habitat preferences of the first hominin groups to disperse out of Africa were variable and complex. However, several environmental variables are suggested to have influenced dispersal routes, including biotic factors such as the presence of carnivores, herbivores, and abiotic factors such as altitude, freshwater availability, temperature, and sources of lithic raw material [132–135]. Overall, patterns of *Homo erectus* dispersal into Asia in the Early and Middle Pleistocene are typically associated with low to mid-altitudes, warm temperatures, and fresh bodies of water. The most important division in Asia may have been latitude, and hominins tended to not live beyond 40 degrees north until ca. 500 Ka [53]. It has also been suggested that *Homo erectus* passively followed herbivore fauna and actively avoided areas populated by carnivores during early dispersals out of Africa [135]. There is also evidence to suggest that *Homo erectus* favored areas nearby sources of good quality material such as flint [135].

Dispersal simulations suggest that the main factors that hominins actively selected for in Eurasian dispersal routes were carnivore avoidance and areas of mixed sedimentary rocks and unconsolidated sediments [135]. The steppe and semi-deserts of Central Asia offered a better fit for these inferred habitat preferences of *Homo erectus* than the adjacent mountain foothills. The lithological surfaces of plains of Central Asia are dominated by mixed sedimentary rock and unconsolidated sediments [136]. Mixed sedimentary rock and unconsolidated sediments are richer in flint cobbles [137], the overwhelming raw material preference of Lower Paleolithic toolmakers in Europe and Asia [138]. This is in contrast to the Central Asia mountain zones which are dominated by more plutonic rocks [136].

The mammal fossil record from Central Asia is relatively sparse, but the limited existing faunal evidence suggests that species of large carnivores are known from the Altai, Tian Shan, and Pamir mountains during the Pleistocene including cave bears, cave lions, hyenas, saber-toothed cats, and various species of wolves [52, 139–141]. If large carnivore avoidance was indeed a driver of *Homo erectus* dispersal routes, it is plausible that the presence of large carnivores, which would have frequented rock shelters and rivers in the mountain foothills, deterred *Homo erectus* from these areas. In contrast, the low-lying region would have fewer large carnivores and would have been more frequented by herbivores. Indeed, it appears that herbivories were distributed over a vast territory across the Turan Depression during the Middle Pleistocene. Faunal material from the Koshkurgan Formation in the first half of the Middle Pleistocene also similar to the contemporaneous faunal complexes known from Europe and Siberia [119, 131]. The continuity in faunal communities across the Central Asian steppe,

which included large herbivores such as forest elephants [119], coupled with access to high-quality raw materials, would have been favorable habitats for the earliest hominins to disperse out of Africa.

In contrast to *Homo erectus*, Neanderthals and Denisovans are known to occupy high-altitude zones, especially mountain regions [142–146]. These hominin taxa frequently occur in the same archaeological units with carnivores and are known to exploit carnivores for their meat and fur [147–150]. Therefore, caves and rock shelters in the Pamir and Altai regions are more consistent with the expected habitats of Denisovans, Neanderthals, and modern humans. This is further supported by the fossil remains of these taxa found in the Inner Asia Mountain Corridor [13, 151].

## Archaeological site preservation in arid Central Asia

The presence of archaeological material throughout the Turan Depression suggests that hominins occupied the Central Asia lowlands periodically in the past. This region provided a habitable environment for hominins and may have been important for early dispersals. However, the long-term climatic record of arid Central Asia is less understood than the adjacent mountain and foothill zones. This is because the plains of arid Central Asia lack organic remains, stratified sequences, datable material, and *in situ* aeolian loess and speleothem deposits. The growing body of paleoclimate work for this region will play an important role by providing more detail on past climate and environment in this poorly understood region during the Middle Pleistocene.

The importance of the plains to early hominin occupation is almost certainly underrepresented in the archaeological record because of poor preservation and lack of stratified sequences. We know relatively little about the occupation of these regions because the majority of archaeological evidence from the steppe and desert consists of surface material [131]. It is possible that arid Central Asia may have been inhabited earlier or more consistently than suggested by the archaeological record, but this is unknown due to lack of evidence. For these reasons, it is especially important to consider all finds, including undated and unstratified material, when assessing patterns of hominin dispersal in this region. Despite preservation limitations, this area is important for understanding early occupation, and in the future, undatable surface material will likely continue to be our most abundant source of information about the occupation of arid Central Asia.

## Conclusion

The existing archaeological evidence suggests that hominins inhabited Central Asia by 0.8 Ma. During this period, the Caspian Sea was at a high stand, indicating a robust supply of freshwater rivers and lakes through the Central Asian plains. The first evidence of hominin occupation is known from the Kopetdag region and the Tian-Shan/ Pamir Mountain foothills where warmer temperatures would have been consistent with conditions of early occupation elsewhere. In comparatively warmer and wetter intervals, such as during super-interglacial MIS 15–13 and MIS 11, hominins could have ventured further north and occupied higher latitudes. The moderate elevation and low frequency of carnivores in the steppe and semi-desert is consistent with the inferred preferences of the earliest hominins to disperse out of Africa.

The example of stalagmite S-12-4 highlights the fact that local and regional conditions did not follow simple long-term trends as could be inferred from low resolution records. Central Asia at the end of MIS 11, before full glacial conditions set in, seems to have witnessed multi-decadal changes in winter moisture supply, likely in part driven by climatic conditions in the North Atlantic realm. Longer speleothem multi-proxy records from Uzbekistan are needed to

evaluate how the region's hydroclimate adopted to the colder glacial conditions of MIS 10. Whether these long-term variations came hand in hand with changes in seasonality (i.e., contrasts between warm/dry and cold/wet season), or climate predictability or volatility [152] remains to be studied in greater detail. Investigation of the seasonally resolved greyscale record from stalagmite S-12-4 using novel methods such as recurrence analysis [153, 154] will allow fresh insights into these questions.

The plains of arid Central Asia have been largely disregarded as a region of significance to early hominin dispersal. This is in part because erosion outpaces sedimentary accumulation in these areas, and therefore stratified and datable material are lacking. However, patterns of lithic scatters reveal that Lower Paleolithic toolmakers inhabited a wide range of steppe, desert, semi-desert areas. Bifaces are present almost exclusively through the plains of Central Asia and at northern latitudes, indicating that this may have been an important zone for early occupation. However, the increasing aridification of Central Asia during the Middle Pleistocene, especially as the Caspian Sea began to experience more consistent low stands over the last 0.5 Ma, may have caused *Homo erectus* populations in Central Asia to become increasingly isolated from other populations [155], possibly leading to declines and regional extinctions. This coincides with the presence of new hominin taxa in Central Asia that were better adapted to mountainous and high-altitude environments. As conditions became more arid and seasonal, the foothills were better suited for consistent occupation and hominins likely opted strictly for the more temperate mountain biomes to the east and south of the Central Asian interior.

## Supporting information

**S1 Fig. Overview image of stalagmite S-12-4 with the areas measured by CRM (A-F).** Blue areas are stitched microscope images obtained with the 20x lens of the CRM and superimposed onto the sample scan. The red rectangles mark areas imaged by CRM. The instrumental settings are identical for all 6 areas and the dimensions are as follows: Area A has a size of 3500 μm x 8000 μm (70000 spectra, distance between measurements 20 μm, integration time 0.1 seconds/measurement). Area B has a size of 3500 μm x 8500 μm (74375 spectra, resolution 20 μm), Area C 3500 x 8200 (71750 spectra, resolution 20 μm), D 3500 μm x 6800 μm (238000 spectra, resolution 10 μm), E 3500 μm x 6800 μm (238000 spectra, resolution 10 μm), and F 1700 μm x 8500 μm (144500 spectra, resolution 10 μm, 0.2 seconds integration time). Area A to E are situated within the sampling trench previously milled for stable isotopes. Area F is on the surface of the sample. Analysis of all Raman spectra measured reveals calcite as the only mineral phase present (see S2 and S3 Figs).
(TIF)

**S2 Fig. Raman spectra of a calcite and aragonite inhouse standards compared to a typical spectrum obtained from S-12-4.** All areas have been identified as calcite and no traces of aragonite have been found.
(TIF)

**S3 Fig. Image based on the intensity distribution of the background fluorescence within area F.** Enhanced fluorescence in Raman spectra of carbonates is often attributable to the presence of organic molecules [4]. This could indicate that the distribution of organic molecules is quantitatively or qualitatively different between darker and the brighter structures.
(TIF)

**S4 Fig. The locations of low and high resolution samples taken from stalagmite S-12-4.**
(TIF)

**S5 Fig. The S-12-4 greyscale profile with an example of layer counting.** A magnified area provides an example of how layers were counted, with peaks labeled with black arrows.
(TIF)

**S6 Fig. Layer counting based chronology with U/Th dates.**
(TIF)

**S1 Table. Paleolithic sites in Central Asia.**
(XLSX)

**S2 Table. $^{230}$Th dating results.** The error is 2s error.
(XLSX)

**S3 Table. Stalagmite S-12-4 stable isotope record.**
(XLSX)

**S4 Table. Stalagmite S-12-4 grey elements.**
(XLSX)

**S1 File. Confocal Raman microscopy methodology.**
(DOCX)

## Acknowledgments

We wish to thank our colleagues Nosir Shukurov and Maksim Petrov (Tashkent) and Alexander Osintsev of the Speleoclub Arabika Irkutsk for their logistical support during the speleological fieldwork. We would also like to thank Evie Grace Merrygold for providing the results of her backward trajectory modelling which we used to describe the climate in the study area. Finally, we would like to thank the three reviewers who provided helpful suggestions and insights to improve this paper.

## Author Contributions

**Conceptualization:** Emma M. Finestone, Paul S. Breeze, Sebastian F. M. Breitenbach, Nick Drake, Farhod Maksudov, Akmal Muhammadiyev, Nicole Boivin, Michael Petraglia.

**Data curation:** Emma M. Finestone, Paul S. Breeze, Sebastian F. M. Breitenbach, Laura Bergmann.

**Formal analysis:** Emma M. Finestone, Paul S. Breeze, Sebastian F. M. Breitenbach, Laura Bergmann, Gernot Nehrke.

**Investigation:** Emma M. Finestone, Paul S. Breeze, Sebastian F. M. Breitenbach, Laura Bergmann, Pete Scott, Yanjun Cai, Arina M. Khatsenovich, Evgeny P. Rybin, Gernot Nehrke.

**Methodology:** Emma M. Finestone, Paul S. Breeze, Sebastian F. M. Breitenbach, Nick Drake, Pete Scott, Yanjun Cai, Michael Petraglia.

**Supervision:** Nick Drake, Farhod Maksudov, Akmal Muhammadiyev, Nicole Boivin, Michael Petraglia.

**Visualization:** Emma M. Finestone, Paul S. Breeze, Sebastian F. M. Breitenbach, Laura Bergmann, Gernot Nehrke.

**Writing – original draft:** Emma M. Finestone, Sebastian F. M. Breitenbach.

**Writing – review & editing:** Emma M. Finestone, Paul S. Breeze, Sebastian F. M. Breitenbach, Nick Drake, Laura Bergmann, Farhod Maksudov, Akmal Muhammadiyev, Pete Scott,

Yanjun Cai, Arina M. Khatsenovich, Evgeny P. Rybin, Gernot Nehrke, Nicole Boivin, Michael Petraglia.

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
