## [Decision Letter · Decision Letter 0]

27 Jun 2022

PONE-D-22-08777Paleolithic occupation of arid Central Asia in the Middle PleistocenePLOS ONE

Dear Dr. Finestone,

Thank you for submitting your manuscript to PLOS ONE. After careful consideration, we feel that it has merit but does not fully meet PLOS ONE’s publication criteria as it currently stands. Therefore, we invite you to submit a revised version of the manuscript that addresses the points raised during the review process. All reviewers agree in thinking that this paper provides an original hypothesis on the spatial and temporal patterning of hominin occupations in Central Asia, based on compilation of the archaeological evidence and palaeoclimatic data sourced from the layer-counting and analysis of a speleotheme. Concerning the latter, R.3 stresses about the factors that could potentially affect the stable oxygen isotope compositions of the stalagmite and invites you to propose an explanation of the reasons why potential driving factors were not considered in the discussion section. About the archaeological framework you provide particularly in the introduction section, R.1 and R2 disagree in some points (hominins first dispersed not Central Asia and some missed sites with bifaces) and recommend some integrations.

We look forward to receiving your revised manuscript.

Kind regards,

Marco Peresani

Academic Editor

PLOS ONE

Journal Requirements:

2.  We note that Figure 1, 3, 4 and 6 in your submission contain copyrighted images. All PLOS content is published under the Creative Commons Attribution License (CC BY 4.0), which means that the manuscript, images, and Supporting Information files will be freely available online, and any third party is permitted to access, download, copy, distribute, and use these materials in any way, even commercially, with proper attribution. For more information, see our copyright guidelines: http://journals.plos.org/plosone/s/licenses-and-copyright.

1. You may seek permission from the original copyright holder of Figure  1, 3, 4 and 6 to publish the content specifically under the CC BY 4.0 license. 

Reviewers' comments:

Reviewer's Responses to Questions

**Comments to the Author**

1. Is the manuscript technically sound, and do the data support the conclusions?

Reviewer #1: Partly

Reviewer #2: Yes

Reviewer #3: Yes

2. Has the statistical analysis been performed appropriately and rigorously? 

Reviewer #1: N/A

Reviewer #2: Yes

Reviewer #3: Yes

3. Have the authors made all data underlying the findings in their manuscript fully available?

Reviewer #1: Yes

Reviewer #2: Yes

Reviewer #3: No

4. Is the manuscript presented in an intelligible fashion and written in standard English?

Reviewer #1: Yes

Reviewer #2: Yes

Reviewer #3: Yes

5. Review Comments to the Author

Reviewer #1: There is much in this paper that I like, and I agree with most of it. I won’t comment on the analysis of the stalagmite as I am not qualified to do so, so I will leave that to a qualified specialist. I don’t mind if you work out my identity as it will be obvious from some of my comments.

My main disagreement is about when hominins first dispersed not Central Asia. You are emphatic that this did not occur until after 1.0 Ma – the age of Kuldara, which is the earliest site in a stratified section. Thus you state (lines 169-170) “the lack of hominin remains or stone tools prior to 1 Ma in Central Asia, suggest that early hominin dispersals into East Asia may have bypassed Central Asia completely”. In your conclusion, you repeat the same (lines 775 ff) ”The existing archaeological evidence suggests that hominins initially bypassed Central Asia, perhaps during periods where climate was unfavorable to occupation during Caspian low stands. Later, a second wave of migration coming out of Africa from west to east, or circling back from east Asia, reached Central Asia by 0.8 Ma”.

I strongly urge caution here: I would not argue that they definitely did disperse across Central Asia in the Early Pleistocene (although I think it likely); rather I suggest that absence of evidence is not the same as evidence of absence. As you state, most of the lower palaeolithic evidence is undated, especially from the western, low-lying part of Central Asia: (line 757) “the plains of arid Central Asia lack organic remains, stratified sequences, datable material, and in situ aeolian loess and speleothem deposits”.

The only region so far explored that contains long stratified sequences in which stone artefacts can be found and dated is the loess-palaeosol area of Tajikistan, that was brilliantly investigated by Andrei Dodonov and Vadim Ranov. I strongly recommend that you consult and include Dodonov’s 2002 monograph “Quaternary of Middle Asia: Stratigraphy, Correlation and Paleogeography. Moscow: Geos. (In Russian)”. Dodonov’s (and others’) research showed that, as in the Chinese Loess Plateau, loess deposition in Tajikistan commenced ca. 2.5 Ma. Unfortunately, few of Dodonov’s profiles exposed the entire sequence (see his monograph). Most of those profiles stopped at palaeosol 11-12 just below the B-M boundary, as at Kuldara. Ranov was lucky as well as very determined in noticing flaked stone at the base of that section and being able to trace it back to source and even conduct a small excavation. What no-one has yet done is to explore systematically the strata below palaeosol 11-12. One problem is that very few of the profiles studied by Dodonov extend into the Early Pleistocene. Another is simply that no-one has looked.

From my own experience in the Chinese Loess Plateau at Gongwangling and especially Shangchen, the first priority is to find a profile that extends back to ca. 2.0 Ma and has extensive and accessible exposures (i.e., not too steep or vegetated). If one is lucky, one can sometimes find flaked stone at the base of a section, and its source might be located by working back upslope (as at Kuldara). Otherwise, searching for stone artefacts in those sections is slow and painstaking. The profiles are normally covered with a thin crust from the previous season’s rainfall and this needs to be flaked off to expose undisturbed loess/palaeosol. It is then a matter of slowly scraping a trowel along the profile, and if lucky, feeling or seeing the edge of a stone. At that point it can be exposed, photographed and removed. I was lucky at Shangchen with the road-side section (see the Nature paper) in that I quickly found three artefacts in palaeosol S27 (ca. 2.0 Ma); my Chinese colleagues then added a small excavation that exposed artefacts ca. 2.1 Ma in age. If I was younger and fitter, and if Tajikistan was accessible, I would love to conduct similar research. My own view is that there should be artefacts in the Tajik profiles that are earlier than Kuldara.

It also seems logical to me that once hominins entered the Palearctic Realm that covers continental Asia and Europe, they would stay within the same type of environment and thus could have dispersed across it to northern China. That, to me at any rate, seems more probable that the scenario you propose, that hominins dispersed into the Oriental Realm of south and southeast Asia, and then entered the Palearctic Realm north of the Qinling Mountains in time to discard artefacts at Shangchen at ca. 2.1 Ma. That route is not only much longer, but would also need to commence well before 2.1 Ma in order to allow time for hominins to colonise successfully the Oriental Realm before they headed north into the Palaearctic Realm.

Additionally, the Early Pleistocene (according to the marine cores and Loess Plateau records) was less severe than the Middle Pleistocene, and surely Central Asia would have contained grasslands and open woodlands as well as potable water after 2.0 Ma? Surely some colonization would have been feasible?

I would therefore suggest that you are less dogmatic over when Central Asia was first colonized. We don’t know, and won’t know until the Early Pleistocene profiles of Tajikistan (and perhaps in other areas of loess deposition) are systematically investigated in the same way as at Shangchen.

Other points

Line 177: Keshafrud is in Iran, not Turkmenistan. Regarding the lithics, I think some of those pieces are geofacts, and Arai and Thibault’s date estimate was just a guess. I would exclude it from Fig. 2 as this estimate is totally unsubstantiated.

Lines 214 ff: the lower palaeolithic sites of Central Asia.

I think you need a brief discussion about the term “site” in this region. Much of the little we know about was collected unsystematically. Are “sites” just places where a few artefacts were found? How many artefacts were collected, and are there any details as to the size and extent of a “site”? Also, with the “pebble culture”: does this simply indicate that nothing else was available – as seems the case at Kuldara, for example – or does it indicate a conscious choice to exclude larger clasts?

Line 278: Hominin remains diagnostic of Homo erectus have been found at sites in neighboring regions, such as Kocabaş, Anatolia, Turkey , Nadaouiyeh Aïn Askar, Syria (73), Gesher Benot Ya’aqov, Palestine (74,75), Zhoukoudian, China (76). You need to add Lantian Gongwangling at 1.63 Ma – see the J. Hum. Evol 2016 paper by Zhu et al..

Line 281: “Homo erectus reached Kocabaş Turkey by ca. 1.1-1.3 Ma (34,79).” This seems a strange statement: reached Turkey from where? Surely by 1.1 – 1.3 Ma, H. erectus was indigenous to western Asia?

Line 622: Re the long interglacials in MIS 15 and MIS 13: “This long period of high humidity has previously been suggested to have facilitated hominin dispersals into Asia (112)”. You can follow up Hao’s suggestion by pointing out that dispersals eastward in this period could also have resulted in the appearance of Acheulean-like bifaces in China.

Figures: the text looked unclear – maybe it was the way it was uploaded? Otherwise, maybe a bolder font?

Fig. 2: I would exclude the Zaisan Basin as none of that material is reliably dated.

Reviewer #2: As a reviewer and as a reader of the journal, I found this article very interesting, especially in the context of the lack of dates for the Lower Palaeolithic of the region. Especially important for me was the fact that I had been working in the region for quite a long time and an "outsider's view" is very important for understanding the processes of human population distribution.

The hypothesis raised in the article is of course very debatable, but I consider this aspect rather a definite plus to start the discussion in the journal.

I would like to point out that I have found a few inaccuracies in the article that need to be corrected and these changes taken into account in the final results of the article. First of all, we are talking about the sites with bifaces, which were not taken into account by the authors.

256 «the time, Vishnyatsky (11) noted that one exception to this trend was Kulbuluk, Tajikistan»

KulbulAk located in Uzbekistan.

256 Vishnyatsky (11) was followed after Kasymov’s habilitation thesis (1990).

At Kulbulak only one handaxe was recovered from Middle Paleolithic layer 5. Other tools Kasymov defined like bifaces from layers 27-28 and according to Vishnyatsky (1996, monography) they looks like Middle Paleolithic tools. From my point of view it is also seems to be Middle Paleolithic.

I would advise the authors to use the following instead of references 71 and 72: Krivoshapkin A, Viola B, Chargynov T, Krajcarz MT, Krajcarz M,Fedorowicz S, Shnaider S, Kolobova K (2020) Middle Paleolithicvariability in Central Asia: lithic assemblage of Sel'Ungur cave.Quat Int 585:88-103. https://doi.org/10.1016/j.quaint.2018.09.051. First of all because this paper is the most recent and is published in English, which makes the data in it more accessible.

300 modern humans. Although no fossils of Paleolithic-age in Central Asia are securely diagnostic of

301 Denisovans or Homo sapiens, both of these taxa are known nearby in Siberia during the Middle

302 and Late Pleistocene (2,91–94)

This is actually not entirely true. The discovery of a human tooth is published in: Kolobova, K., Flas, D., Derevianko, A.P., Pavlenok, K., Islamov, U.I., Krivoshapkin, A.I.,2013. The Kulbulak bladelet tradition in the upper paleolithic of central Asia.Archaeol. Ethnol. Anthropol. Eurasia 41, 2–25. https://doi.org/10.1016/j.aeae.2013.11.002. «In 2009, the bottom of horizon 2.1 yielded a tooth, which, according to B. Viola’s personal communication (2009), is a third lower premolar of Homo sapiens. The tooth was found in an undisturbed stratigraphic context and is well preserved”.

270 initially classified as Homo erectus (69), however, this identification was later re-evaluated and

271 the taxonomic attribution is now considered indeterminate (13,70).

Actually, this is not the most accurate published data. Here is the last one: “but several morphological details contradict this interpretation, and indicate that at least five of the teeth do not represent hominins (Viola, 2009; Viola and Krivoshapkin, 2014; contra Zubov, 2009). The juvenile humerus on the other hand is clearly hominin. It preserves most of the shaft from the distal epiphyseal line to the proximal part of the deltoid tuberosity, and seems very long and gracile, though with very thick cortical bone. The distal half of the shaft is triangular in cross section and flattened mediolaterally, reminiscent of the morphology seen in Neanderthals, but also other archaic hominins (Viola, pers. obs.)” Krivoshapkin A, Viola B, Chargynov T, Krajcarz MT, Krajcarz M,Fedorowicz S, Shnaider S, Kolobova K (2020) Middle Paleolithicvariability in Central Asia: lithic assemblage of Sel'Ungur cave.Quat Int 585:88-103. https://doi.org/10.1016/j.quaint.2018.09.051

303 Middle and Late Pleistocene (92,93,95–98), and have been uncovered in Uzbekistan at Teshik

304 Tash (99) and Obi-Rakhmat Grotto (100) alongside Middle Paleolithic assemblages.

Here the authors should mention a bone from Sel'Ungur, anthropological remains from Angillac Cave by Glantz M and a tooth from the Khudzhi site by Ranov V in Tadzhikistan.

324 The geographic locations of 132 Upper Paleolithic, Middle Paleolithic and Lower

325 Paleolithic assemblages were collected from the published literature (Supplementary Table 1)

For the reference to the Shugnou site, I would ask to use the original publication with the dates: Ranov, V.A., Kolobova, K.A., Krivoshapkin, A.I., 2012. The Upper Paleolithic assemblagesof Shugnou, Tajikistan. Archaeol. Ethnol. Anthropol. Eurasia 40 (2), 2–24.

326 We broadly attributed sites to the Lower Paleolithic, Middle Paleolithic, and/or Upper

327 Paleolithic, and noted the presence/absence of bifaces.

Here, the Middle Palaeolithic site with bifaces is definitely not on the list: Gofilabad site from- Khujageldiev, T., Kolobova, K., Shnaider, S., Krivoshapkin, A. 2019. The first evidence of bifacial technology in the middle palaeolithic of Tajikistan. Stratum Plus, 265–277.

Supplimentary table 121 - Sel’ungur Cave – In this Cave several plano-convex bifaces have been found. Krivoshapkin A, Viola B, Chargynov T, Krajcarz MT, Krajcarz M,Fedorowicz S, Shnaider S, Kolobova K (2020) Middle Paleolithicvariability in Central Asia: lithic assemblage of Sel'Ungur cave.Quat Int 585:88-103.

I would like to point out to the editors that the article is very interesting and has very significant citation potential.

Kseniya Kolobova

Reviewer #3: Central Asia is a key region for studying hominin dispersal and its relationship with arid climate changes in the Asia interior during the Pleistocene period. However, the archaeological works are relative scattered and the high-resolution hydroclimate records are rare in this remote region. This paper present new stratified lithic assemblages from Central Asia by reviewing the published literatures, which provides simple but important framework for understanding the spatial and temporal patterning of hominin occupations in central Asia. This paper also presents new speleothem record from the Amir Timur Cave in southern Uzbekistan, which shed light on high-resolution hydroclimate changes in central Asia between ~387 and ~405 ka BP. Overall, this paper is well written and the data reported are important, thus I feel that this is a good contribution to PLOS ONE and the manuscript can be accepted after minor revisions.

Major comments:

1. The authors interpret the reduced winter precipitation relative to the overall annual budget would be reflected in higher δ18O values of the stalagmite (Lines 665-666). This is an interesting interpretation. However, there are many factors could potentially affect the stable oxygen isotope compositions of the stalagmite, such as moisture source dynamics, seasonality of precipitation, amount of precipitation, and temperature. It is not clear why the other potential driving factors were excluded. I noticed that the authors shortly discussed in the supplementary text, but I do feel this part (S1 text) should be moved to the main text and more evidence or citations should be involved to strengthen the arguments.

2. The Conclusion Section seems a little bit long. I suggest shortening this part by focuses on the main findings of this work.

3. The PLOS Data policy requires authors to make all data underlying the findings described in their manuscript fully available without restriction, however, I cannot find the stable isotope data of the stalagmite reported in this work. Please double check if these data have to be attached in the supplementary materials.

Specific comments:

Line 93: add “by” before “the Ural Mountains”.

Line 96: “Central Asia study region” looks strange. Please rewrite.

Line 105: add “,” after “Consequently”.

Line 135: change “and mid-altitude of Central Asia” to “in the mid-altitude of Central Asia”?

Lines 159-161: Please rewrite this sentence.

Lines 166-170: This sentence is too long. Please rewrite.

Lines 262-264: This sentence is difficult to follow. Please rewrite.

Line 414: change to “Assemblage of Geographic Distribution”?

Line 546: add “,” after “period”

Line 649: add “,” after “climate”

Figure 1: add “N” and “E” after the coordinate numbers

6. PLOS authors have the option to publish the peer review history of their article (what does this mean?). If published, this will include your full peer review and any attached files.

Reviewer #1: No

Reviewer #2: **Yes: **Kseniya Kolobova

Reviewer #3: No

---

## [Author Response · Author response to Decision Letter 0]

9 Aug 2022

PONE-D-22-08777

Paleolithic occupation of arid Central Asia in the Middle Pleistocene

PLOS ONE

Dear Dr. Peresani,

Thank you for your consideration of Paleolithic occupation of arid Central Asia in the Middle Pleistocene. We submit our revised manuscript for further consideration. We appreciate the comments received from our three reviewers. They recognized the interest of our findings to PLOS ONE readers and made suggestions where our text needed clarification. We agree with their assessment of this paper and have addressed all of their comments in our revised draft. In the following section we’ve listed each reviewer’s comments and summarized the actions we took in response (in bold). 

Journal Requirements:

Thank you for drawing attention to this. We have now named our files in accordance with the PLOS ONE style requirements.

2. We note that Figure 1, 3, 4 and 6 in your submission contain copyrighted images. All PLOS content is published under the Creative Commons Attribution License (CC BY 4.0), which means that the manuscript, images, and Supporting Information files will be freely available online, and any third party is permitted to access, download, copy, distribute, and use these materials in any way, even commercially, with proper attribution. For more information, see our copyright guidelines: http://journals.plos.org/plosone/s/licenses-and-copyright.

1. You may seek permission from the original copyright holder of Figure 1, 3, 4 and 6 to publish the content specifically under the CC BY 4.0 license. 

All maps were produced by one of the authors (PB). The basemap images that are used in figures 1 and 4 were produced by the author in a GIS, derived from freely available unsymbolised raw data (SRTM and Worldclim). The basemaps in figures 3 and 6 are the Natural earth data, which is in the public domain [https://www.naturalearthdata.com/about/terms-of-use/ No permission is needed to use Natural Earth. Crediting the authors is unnecessary] and freely available for use, without credit required. We have now added clarifications to the captions as to the data sources used by the authors to produce the maps.

We have included all of our data in the supplementary material, including new supplementary tables (S3 and S4 tables).

We have added the following references to our list in order to cite the data that was used to generate maps: 

• Fick, S.E. and R.J. Hijmans, 2017. WorldClim 2: new 1km spatial resolution climate surfaces for global land areas. International Journal of Climatology 37 (12): 4302-4315.

• Jarvis, A., Reuter, H.I., Nelson, A., Guevara, E., 2008. Hole-filled seamless SRTM data V4, International Centre for Tropical Agriculture (CIAT) [WWW Document]. URL http://srtm.csi.cgiar.org

• Lehner, B., Grill, G., 2013. Global river hydrography and network routing: baseline data and new approaches to study the world’s large river systems. Hydrological Processes 27, 2171–2186. doi:10.1002/hyp.9740

• Pekel, J.-F., Cottam, A., Gorelick, N., Belward, A.S., 2016. High-resolution mapping of global surface water and its long-term changes. Nature 540, 418–422. doi:10.1038/nature20584

We have also added the following references to our list per reviewer 2’s suggestions:

• Kasymov MR. Problemy paleolita Srednei Azii i Yuzhnogo Kazakhstana (po materialam mnogosloinoi paleoliticheskoi stoianki Kulbulak). Abstract of History Dr Diss Novosibirsk. 1990

• Vishnyatsky LB. Paleolit Srednei Azii i Kazakhstana (The Paleolithic of Central Asia and Kazakhstan), Evropeiskiy Dom. St Petersburg. 1996.

• Glantz M, Viola B, Wrinn P, Chikisheva T, Derevianko A, Krivoshapkin A, et al. New hominin remains from Uzbekistan. Journal of Human Evolution. 2008;55(2):223–37.

• Trinkaus E, Ranov VA, Lauklin S. Middle Paleolithic human deciduous incisor from Khudji, Tajikistan. Journal of Human Evolution. 2000;38(4):575–84. 

• Ranov VA, Schafer J. The Palaeolithic of the late middle Pleistocene in central Asia, 400-100 ka ago. BAR International Series. 2000;850:77–92.

• Krivoshapkin A, Viola B, Chargynov T, Krajcarz MT, Krajcarz M, Fedorowicz S, et al. Middle Paleolithic variability in Central Asia: lithic assemblage of Sel’Ungur cave. Quaternary International. 2020;535:88–103. 

We have added the following references to our reference list per reviewer 3’s suggestion to expand the details of our interpretation that reduced winter precipitation relative to the overall annual budget would be reflected in higher δ18O values of the stalagmite:

• Breitenbach SF, Adkins JF, Meyer H, Marwan N, Kumar KK, Haug GH. Strong influence of water vapor source dynamics on stable isotopes in precipitation observed in Southern Meghalaya, NE India. Earth and Planetary Science Letters. 2010;292(1–2):212–20. 

• Lechleitner FA, Baldini JUL, Breitenbach SFM, Fohlmeister J, McIntyre C, Goswami B, et al. Hydrological and climatological controls on radiocarbon concentrations in a tropical stalagmite. Geochimica et Cosmochimica Acta [Internet]. 2016 Dec 1;194:233–52. Available from: https://www.sciencedirect.com/science/article/pii/S0016703716304999

• Nehrke G, Nouet J. Confocal Raman microscope mapping as a tool to describe different mineral and organic phases at high spatial resolution within marine biogenic carbonates: case study on Nerita undata (Gastropoda, Neritopsina). Biogeosciences. 2011;8(12):3761-9. doi: 10.5194/bg-8-3761-2011.

• Milano S, Nehrke G. Microstructures in relation to temperature-induced aragonite-to-calcite transformation in the marine gastropod Phorcus turbinatus. PLOS ONE. 2018;13(10):e0204577. doi: 10.1371/journal.pone.0204577

• Ramesh K, Melzner F, Griffith AW, Gobler CJ, Rouger C, Tasdemir D, et al. In vivo characterization of bivalve larval shells: a confocal Raman microscopy study. Journal of The Royal Society Interface. 2018;15(141).

• Wall M, Nehrke G. Reconstructing skeletal fiber arrangement and growth mode in the coral Porites lutea (Cnidaria, Scleractinia): a confocal Raman microscopy study. Biogeosciences. 2012;9(11):4885-95. doi: 10.5194/bg-9-4885-2012.

Reviewer #1:

There is much in this paper that I like, and I agree with most of it. I won’t comment on the analysis of the stalagmite as I am not qualified to do so, so I will leave that to a qualified specialist. I don’t mind if you work out my identity as it will be obvious from some of my comments.

My main disagreement is about when hominins first dispersed not Central Asia. You are emphatic that this did not occur until after 1.0 Ma – the age of Kuldara, which is the earliest site in a stratified section. Thus you state (lines 169-170) “the lack of hominin remains or stone tools prior to 1 Ma in Central Asia, suggest that early hominin dispersals into East Asia may have bypassed Central Asia completely”. In your conclusion, you repeat the same (lines 775 ff) ”The existing archaeological evidence suggests that hominins initially bypassed Central Asia, perhaps during periods where climate was unfavorable to occupation during Caspian low stands. Later, a second wave of migration coming out of Africa from west to east, or circling back from east Asia, reached Central Asia by 0.8 Ma”.

I strongly urge caution here: I would not argue that they definitely did disperse across Central Asia in the Early Pleistocene (although I think it likely); rather I suggest that absence of evidence is not the same as evidence of absence. As you state, most of the lower palaeolithic evidence is undated, especially from the western, low-lying part of Central Asia: (line 757) “the plains of arid Central Asia lack organic remains, stratified sequences, datable material, and in situ aeolian loess and speleothem deposits”.

The only region so far explored that contains long stratified sequences in which stone artefacts can be found and dated is the loess-palaeosol area of Tajikistan, that was brilliantly investigated by Andrei Dodonov and Vadim Ranov. I strongly recommend that you consult and include Dodonov’s 2002 monograph “Quaternary of Middle Asia: Stratigraphy, Correlation and Paleogeography. Moscow: Geos. (In Russian)”. Dodonov’s (and others’) research showed that, as in the Chinese Loess Plateau, loess deposition in Tajikistan commenced ca. 2.5 Ma. Unfortunately, few of Dodonov’s profiles exposed the entire sequence (see his monograph). Most of those profiles stopped at palaeosol 11-12 just below the B-M boundary, as at Kuldara. Ranov was lucky as well as very determined in noticing flaked stone at the base of that section and being able to trace it back to source and even conduct a small excavation. What no-one has yet done is to explore systematically the strata below palaeosol 11-12. One problem is that very few of the profiles studied by Dodonov extend into the Early Pleistocene. Another is simply that no-one has looked.

From my own experience in the Chinese Loess Plateau at Gongwangling and especially Shangchen, the first priority is to find a profile that extends back to ca. 2.0 Ma and has extensive and accessible exposures (i.e., not too steep or vegetated). If one is lucky, one can sometimes find flaked stone at the base of a section, and its source might be located by working back upslope (as at Kuldara). Otherwise, searching for stone artefacts in those sections is slow and painstaking. The profiles are normally covered with a thin crust from the previous season’s rainfall and this needs to be flaked off to expose undisturbed loess/palaeosol. It is then a matter of slowly scraping a trowel along the profile, and if lucky, feeling or seeing the edge of a stone. At that point it can be exposed, photographed and removed. I was lucky at Shangchen with the road-side section (see the Nature paper) in that I quickly found three artefacts in palaeosol S27 (ca. 2.0 Ma); my Chinese colleagues then added a small excavation that exposed artefacts ca. 2.1 Ma in age. If I was younger and fitter, and if Tajikistan was accessible, I would love to conduct similar research. My own view is that there should be artefacts in the Tajik profiles that are earlier than Kuldara.

It also seems logical to me that once hominins entered the Palearctic Realm that covers continental Asia and Europe, they would stay within the same type of environment and thus could have dispersed across it to northern China. That, to me at any rate, seems more probable that the scenario you propose, that hominins dispersed into the Oriental Realm of south and southeast Asia, and then entered the Palearctic Realm north of the Qinling Mountains in time to discard artefacts at Shangchen at ca. 2.1 Ma. That route is not only much longer, but would also need to commence well before 2.1 Ma in order to allow time for hominins to colonise successfully the Oriental Realm before they headed north into the Palaearctic Realm.

Additionally, the Early Pleistocene (according to the marine cores and Loess Plateau records) was less severe than the Middle Pleistocene, and surely Central Asia would have contained grasslands and open woodlands as well as potable water after 2.0 Ma? Surely some colonization would have been feasible?

I would therefore suggest that you are less dogmatic over when Central Asia was first colonized. We don’t know, and won’t know until the Early Pleistocene profiles of Tajikistan (and perhaps in other areas of loess deposition) are systematically investigated in the same way as at Shangchen.

We appreciate these insights and agree with them. Although our paper primarily addresses hominin dispersal in the Middle Pleistocene, we want to make it clear that it is possible that the first occupations occurred in earlier intervals. We agree that absence of archaeological material in the Early Pleistocene does not equate to absence of hominins, especially given the lack of systematic investigation of this time interval in comparison to other regions. We have modified our text so as not imply that the earliest archaeological sites necessarily represent the earliest dispersals. For example, we have edited the section on “the timing of the earliest occupation of Central Asia” to read:

“The lack of hominin remains or stone tools prior to 1 Ma in Central Asia may suggest that early hominin dispersals into East Asia bypassed Central Asia completely and later entered Central Asia either through an east-to-west route, or in subsequent migration events from Africa north-eastward. However, Early Pleistocene stratified deposits are sparce and largely unexplored in Central Asia. It is possible that hominins initially colonialized and dispersed across Central Asia during Early Pleistocene migrations to northern China, but that evidence is currently lacking because of an absence of systematic investigations of this time interval.”

Other points

Line 177: Keshafrud is in Iran, not Turkmenistan. Regarding the lithics, I think some of those pieces are geofacts, and Arai and Thibault’s date estimate was just a guess. I would exclude it from Fig. 2 as this estimate is totally unsubstantiated.

Thank you for this comment. We have corrected this in the text and removed all unsubstantiated date estimates from Figure 2.

Lines 214 ff: the lower palaeolithic sites of Central Asia.

I think you need a brief discussion about the term “site” in this region. Much of the little we know about was collected unsystematically. Are “sites” just places where a few artefacts were found? How many artefacts were collected, and are there any details as to the size and extent of a “site”? Also, with the “pebble culture”: does this simply indicate that nothing else was available – as seems the case at Kuldara, for example – or does it indicate a conscious choice to exclude larger clasts?

We agree that the term “site” in this manuscript needs a brief definition. In Central Asia, many lithics were historically collected unsystematically and the sample sizes are highly variable. We have clarified in the text that: “We define a “site” as any place where paleolithic stone tools were recovered.”

Regarding “pebble culture” –we agree that “pebble culture” could be a product of both local raw material availability or hominin preference. However, we do not think a discussion of these contributing factors is relevant to our current manuscript. We mention “pebble culture” only within a brief summary of how Lower Paleolithic assemblages and typologies that have been categorized and identified in the literature.

Line 278: Hominin remains diagnostic of Homo erectus have been found at sites in neighboring regions, such as Kocabaş, Anatolia, Turkey , Nadaouiyeh Aïn Askar, Syria (73), Gesher Benot Ya’aqov, Palestine (74,75), Zhoukoudian, China (76). You need to add Lantian Gongwangling at 1.63 Ma – see the J. Hum. Evol 2016 paper by Zhu et al..

Thank you for pointing this out. We have added Lantian Gongwangling.

Line 281: “Homo erectus reached Kocabaş Turkey by ca. 1.1-1.3 Ma (34,79).” This seems a strange statement: reached Turkey from where? Surely by 1.1 – 1.3 Ma, H. erectus was indigenous to western Asia?

We agree with this comment and have deleted this sentence. 

Line 622: Re the long interglacials in MIS 15 and MIS 13: “This long period of high humidity has previously been suggested to have facilitated hominin dispersals into Asia (112)”. You can follow up Hao’s suggestion by pointing out that dispersals eastward in this period could also have resulted in the appearance of Acheulean-like bifaces in China.

Thank you for this suggestion. We have mentioned the appearance of Acheulean-like bifaces in China as a supporting point. 

Figures: the text looked unclear – maybe it was the way it was uploaded? Otherwise, maybe a bolder font?

We have changed the font on the map figures. The high resolution uploads should also be clear.

Fig. 2: I would exclude the Zaisan Basin as none of that material is reliably dated.

We removed all unsubstantiated date estimates from Figure 2.

Reviewer #2:

As a reviewer and as a reader of the journal, I found this article very interesting, especially in the context of the lack of dates for the Lower Palaeolithic of the region. Especially important for me was the fact that I had been working in the region for quite a long time and an "outsider's view" is very important for understanding the processes of human population distribution.

The hypothesis raised in the article is of course very debatable, but I consider this aspect rather a definite plus to start the discussion in the journal.

I would like to point out that I have found a few inaccuracies in the article that need to be corrected and these changes taken into account in the final results of the article. First of all, we are talking about the sites with bifaces, which were not taken into account by the authors.

256 «the time, Vishnyatsky (11) noted that one exception to this trend was Kulbuluk, Tajikistan»

KulbulAk located in Uzbekistan.

256 Vishnyatsky (11) was followed after Kasymov’s habilitation thesis (1990).

At Kulbulak only one handaxe was recovered from Middle Paleolithic layer 5. Other tools Kasymov defined like bifaces from layers 27-28 and according to Vishnyatsky (1996, monography) they looks like Middle Paleolithic tools. From my point of view it is also seems to be Middle Paleolithic.

Thank you for pointing this out. We have changed the text to reflect this: 

“At the time, Vishnyatsky (11) noted that one possible exception to this trend was Kulbulak, Uzbekistan, where handaxes and bifaces appear to be present in mountain foothills and at a more southern latitude than expected (67). However, according to Vishnyatsky (68) these bifaces are more likely Middle Paleolithic tools and more recently, Kolobova et al. (69) re-evaluated the Kulbulak assemblage and concluded that Acheulean tools are absent at Kulbulak altogether.”

I would advise the authors to use the following instead of references 71 and 72: Krivoshapkin A, Viola B, Chargynov T, Krajcarz MT, Krajcarz M,Fedorowicz S, Shnaider S, Kolobova K (2020) Middle Paleolithicvariability in Central Asia: lithic assemblage of Sel'Ungur cave.Quat Int 585:88-103. https://doi.org/10.1016/j.quaint.2018.09.051. First of all because this paper is the most recent and is published in English, which makes the data in it more accessible.

Thank you. We have added this citation instead.

300 modern humans. Although no fossils of Paleolithic-age in Central Asia are securely diagnostic of

301 Denisovans or Homo sapiens, both of these taxa are known nearby in Siberia during the Middle

302 and Late Pleistocene (2,91–94)

This is actually not entirely true. The discovery of a human tooth is published in: Kolobova, K., Flas, D., Derevianko, A.P., Pavlenok, K., Islamov, U.I., Krivoshapkin, A.I.,2013. The Kulbulak bladelet tradition in the upper paleolithic of central Asia.Archaeol. Ethnol. Anthropol. Eurasia 41, 2–25. https://doi.org/10.1016/j.aeae.2013.11.002. «In 2009, the bottom of horizon 2.1 yielded a tooth, which, according to B. Viola’s personal communication (2009), is a third lower premolar of Homo sapiens. The tooth was found in an undisturbed stratigraphic context and is well preserved”.

Thank you. We have added the following sentence: “A lower premolar attributed to Homo sapiens has been recovered from Kulbulak, Uzbekistan (95)”

270 initially classified as Homo erectus (69), however, this identification was later re-evaluated and

271 the taxonomic attribution is now considered indeterminate (13,70).

Actually, this is not the most accurate published data. Here is the last one: “but several morphological details contradict this interpretation, and indicate that at least five of the teeth do not represent hominins (Viola, 2009; Viola and Krivoshapkin, 2014; contra Zubov, 2009). The juvenile humerus on the other hand is clearly hominin. It preserves most of the shaft from the distal epiphyseal line to the proximal part of the deltoid tuberosity, and seems very long and gracile, though with very thick cortical bone. The distal half of the shaft is triangular in cross section and flattened mediolaterally, reminiscent of the morphology seen in Neanderthals, but also other archaic hominins (Viola, pers. obs.)” Krivoshapkin A, Viola B, Chargynov T, Krajcarz MT, Krajcarz M,Fedorowicz S, Shnaider S, Kolobova K (2020) Middle Paleolithic variability in Central Asia: lithic assemblage of Sel'Ungur cave.Quat Int 585:88-103. https://doi.org/10.1016/j.quaint.2018.09.051

“Hominin remains recovered from Sel’Ungur were initially classified as Homo erectus (74). This identification was later re-evaluated and the taxonomic attribution is now considered indeterminate (13,75) and features of a juvenile humerus from Sel’Ungur are similar to the morphology of Neanderthals or other archaic hominins (76).”

303 Middle and Late Pleistocene (92,93,95–98), and have been uncovered in Uzbekistan at Teshik

304 Tash (99) and Obi-Rakhmat Grotto (100) alongside Middle Paleolithic assemblages.

Here the authors should mention a bone from Sel'Ungur, anthropological remains from Angillac Cave by Glantz M and a tooth from the Khudzhi site by Ranov V in Tadzhikistan.

Thank you. We have changed this to read:

“Neanderthal remains are also present in Siberia during the Middle and Late Pleistocene (97,98,100–103), and have been uncovered in Uzbekistan at Teshik-Tash (104) and Obi-Rakhmat Grotto (105) alongside Middle Paleolithic assemblages. Hominin remains with morphological affinities to Neanderthals have also been uncovered at Sel’ungur (76), Anghilak Cave (6) and Khudji (106,107).”

324 The geographic locations of 132 Upper Paleolithic, Middle Paleolithic and Lower

325 Paleolithic assemblages were collected from the published literature (Supplementary Table 1)

For the reference to the Shugnou site, I would ask to use the original publication with the dates: Ranov, V.A., Kolobova, K.A., Krivoshapkin, A.I., 2012. The Upper Paleolithic assemblages of Shugnou, Tajikistan. Archaeol. Ethnol. Anthropol. Eurasia 40 (2), 2–24.

We have fixed this. Thank you.

326 We broadly attributed sites to the Lower Paleolithic, Middle Paleolithic, and/or Upper

327 Paleolithic, and noted the presence/absence of bifaces.

Here, the Middle Palaeolithic site with bifaces is definitely not on the list: Gofilabad site from- Khujageldiev, T., Kolobova, K., Shnaider, S., Krivoshapkin, A. 2019. The first evidence of bifacial technology in the middle palaeolithic of Tajikistan. Stratum Plus, 265–277.

Supplimentary table 121 - Sel’ungur Cave – In this Cave several plano-convex bifaces have been found. Krivoshapkin A, Viola B, Chargynov T, Krajcarz MT, Krajcarz M,Fedorowicz S, Shnaider S, Kolobova K (2020) Middle Paleolithicvariability in Central Asia: lithic assemblage of Sel'Ungur cave.Quat Int 585:88-103.

We have added these sites to the sample of bifaces reported in 1) Supplementary Table 1, 2) Figure 4, and 3) the statistical comparison of elevation and latitude between bifaces and other paleolithic assemblages. 

I would like to point out to the editors that the article is very interesting and has very significant citation potential.

Kseniya Kolobova

Reviewer #3:

 Central Asia is a key region for studying hominin dispersal and its relationship with arid climate changes in the Asia interior during the Pleistocene period. However, the archaeological works are relative scattered and the high-resolution hydroclimate records are rare in this remote region. This paper present new stratified lithic assemblages from Central Asia by reviewing the published literatures, which provides simple but important framework for understanding the spatial and temporal patterning of hominin occupations in central Asia. This paper also presents new speleothem record from the Amir Timur Cave in southern Uzbekistan, which shed light on high-resolution hydroclimate changes in central Asia between ~387 and ~405 ka BP. Overall, this paper is well written and the data reported are important, thus I feel that this is a good contribution to PLOS ONE and the manuscript can be accepted after minor revisions.

Major comments:

1. The authors interpret the reduced winter precipitation relative to the overall annual budget would be reflected in higher δ18O values of the stalagmite (Lines 665-666). This is an interesting interpretation. However, there are many factors could potentially affect the stable oxygen isotope compositions of the stalagmite, such as moisture source dynamics, seasonality of precipitation, amount of precipitation, and temperature. It is not clear why the other potential driving factors were excluded. I noticed that the authors shortly discussed in the supplementary text, but I do feel this part (S1 text) should be moved to the main text and more evidence or citations should be involved to strengthen the arguments.

It is true that several factors might (and likely do) influence the d18O signal that is recorded in the stalagmite. Possible mechanisms include i) moisture history (including changes of source and distance to the source(s)), ii) amount effect, iii) seasonal distribution of precipitation, iv) prior carbonate precipitation, v) environmental temperature, vi) evaporation dynamics within the cave, and vii) mineralogy (calcite vs aragonite). In our case (Amir Timur) we find that the most likely scenario is a change in seasonal effective moisture reaching the cave. In the revised manuscript we have moved the SOM discussion to the main text and provide a detailed discussion on each of the possible influencing factors and processes that can affect stalagmite d18O. We also included additional data that has been measured since submission. The relative importance of each of these factors controls the final d18O signal we see in the speleothem. To prove the absence of mineralogical changes, we added detailed confocal Raman microscopy maps that show that only calcite is present as mineral phase in this stalagmite. The additional analyses were conducted by (new coauthor) Gernot Nehrke at the AWI Bremerhaven who is a leading CRM expert.

2. The Conclusion Section seems a little bit long. I suggest shortening this part by focuses on the main findings of this work.

In accordance with Reviewer #1’s comment about being careful to make assumptions about when hominins first dispersed into Central Asia, we have taken out the first few sentences of the conclusion about early dispersal routes.

3. The PLOS Data policy requires authors to make all data underlying the findings described in their manuscript fully available without restriction, however, I cannot find the stable isotope data of the stalagmite reported in this work. Please double check if these data have to be attached in the supplementary materials.

We have attached the stable isotope record as supplementary table 3 and the grey values as supplementary table 4.

Specific comments:

Line 93: add “by” before “the Ural Mountains”.

Line 96: “Central Asia study region” looks strange. Please rewrite.

Line 105: add “,” after “Consequently”.

Line 135: change “and mid-altitude of Central Asia” to “in the mid-altitude of Central Asia”?

Lines 159-161: Please rewrite this sentence.

Lines 166-170: This sentence is too long. Please rewrite.

Lines 262-264: This sentence is difficult to follow. Please rewrite.

Line 414: change to “Assemblage of Geographic Distribution”?

Line 546: add “,” after “period”

Line 649: add “,” after “climate”

Figure 1: add “N” and “E” after the coordinate numbers

Thank you for pointing out these errors. We have fixed them each in the resubmission.

We believe we have substantially addressed the comments of all three reviewers. Thank you for your consideration and we look forward to hearing from you. 

Sincerely,

Emma M. Finestone

Assistant Curator of Human Origins, The Cleveland Museum of Natural History

Research Associate, The Max Planck Institute for the Science of Human History

---

## [Editor Report · Decision Letter 1]

19 Aug 2022

Paleolithic occupation of arid Central Asia in the Middle Pleistocene

PONE-D-22-08777R1

Dear Dr. Finestone,

We’re pleased to inform you that your manuscript has been judged scientifically suitable for publication after cheking of all interventions made following the reviewers' recommendations and remarks. Corrections, integrations, deleting sentences and adding of references have been fixed. I've only one concern about the definition of site expressed in line 325. Given the impact of postdepositional processes and dispersion of stone artefacts in sediments of different nature, I find this definition too generic and suggest to add the following at the end of the sentence: ...recovered in primary or sub-primary position. In case, you can fix this intervention during proofs cheking.

Note also that your manuscript will be formally accepted for publication once it meets all outstanding technical requirements.

Kind regards,

Marco Peresani

Academic Editor

PLOS ONE

---

## [Editor Report · Acceptance letter]

13 Oct 2022

PONE-D-22-08777R1 

Paleolithic occupation of arid Central Asia in the Middle Pleistocene 

Dear Dr. Finestone:

I'm pleased to inform you that your manuscript has been deemed suitable for publication in PLOS ONE. Congratulations! Your manuscript is now with our production department. 

Kind regards, 

on behalf of

Dr. Marco Peresani 

Academic Editor

PLOS ONE